# Statistical forecast of seasonal discharge in Central Asia using observational records: development of a generic linear modelling tool for operational water resource management

Heiko Apel[1], Zharkinay Abdykerimova[2], Marina Agalhanova[3], Azamat Baimaganbetov[4], Nadejda Gavrilenko[5], Lars Gerlitz[1], Olga Kalashnikova[6], Katy Unger-Shayesteh[1], Sergiy Vorogushyn[1], Abror Gafurov[1]

[1]GFZ German Research Centre for Geoscience, Section 5.4 Hydrology, Potsdam, Germany

[2]Hydro-Meteorological Service of Kyrgyzstan, Bishkek, Kyrgyzstan

[3]Hydro-Meteorological Service of Turkmenistan, Ashgabat, Turkmenistan

[4]Hydro-Meteorological Service of Kazakhstan, Almaty, Kazakhstan

[5]Hydro-Meteorological Service of Uzbekistan, Tashkent, Uzbekistan

[6]CAIAG Central Asian Institute for Applied Geoscience, Bishkek, Kyrgyzstan

*Correspondence to*: Heiko Apel (heiko.apel@gfz-potsdam.de)

**Abstract.** The semi-arid regions of Central Asia crucially depend on the water resources supplied by the mountainous areas of the Tien Shan, Pamir and Altai mountains. During the summer months the snow and glacier melt dominated river discharge originating in the mountains provides the main water resource available for agricultural production, but also for storage in reservoirs for energy generation during the winter months. Thus a reliable seasonal forecast of the water resources is crucial for a sustainable management and planning of water resources. In fact, seasonal forecasts are mandatory tasks of all national hydro-meteorological services in the region. In order to support the operational seasonal forecast procedures of hydro-meteorological services, this study aims at the development of a generic tool for deriving statistical forecast models of seasonal river discharge based solely on observational records. The generic model structure is kept as simple as possible in order to be driven by meteorological and hydrological data readily available at the hydro-meteorological services, and to be applicable for all catchments in the region. As snowmelt dominates summer runoff, the main meteorological predictors for the forecast models are monthly values of winter precipitation and temperature, satellite based snow cover data, and antecedent discharge. This basic predictor set was further extended by multi-monthly means of the individual predictors, as well as composites of the predictors. Forecast models are derived based on these predictors as linear combinations of up to 4 predictors. A user selectable number of best models is extracted automatically by the developed model fitting algorithm, which includes a test for robustness by a leave-one-out cross validation. Based on the cross validation the predictive uncertainty was quantified for every prediction model. Forecasts of the mean seasonal discharge of the period April to September are derived every month starting from January until June. The application of the model for several catchments in Central Asia - ranging from small to

the largest rivers (240 km$^2$ to 290,000 km$^2$ catchment area) – for the period 2000-2015 provided skilful forecasts for most catchments already in January, with adjusted R$^2$ values of the best model in the range of 0.6 – 0.8 for most of the catchments. The skill of the prediction increased every following month, i.e. with reduced lead time, with adjusted R$^2$ values usually in the range 0.8 – 0.9 for the best and 0.7 – 0.8 on average of the set of models in April just before the prediction period. The later forecasts in May and June improve further due to the high predictive power of the discharge in the first two months of the snowmelt period. The improved skill of the set of forecast models with decreasing lead time resulted in narrow predictive uncertainty bands at the beginning of the snowmelt period. In summary, the proposed generic automatic forecast model development tool provides robust predictions for seasonal water availability in Central Asia, which will be tested against the official forecasts in the upcoming years, with the vision of operational implementation.

## 1 Introduction

The Central Asian (CA) region encompassing the five countries Kazakhstan, Kyrgyzstan, Tajikistan, Turkmenistan and Uzbekistan as well as northern parts of Afghanistan and north-western regions of China is characterized by the presence of two major mountain systems. The Tien Shan and Pamir mountains are drained by a number of endorheic river systems such as Amudarya, Syrdarya, Ili, Tarim and a few smaller ones. The Central Asian river basins are characterized by semi-arid climate with strong seasonal variation of precipitation. Most precipitation falls as snow during winter and spring months in Western and Northern Tien Shan (Aizen et al., 1995, 1996;Sorg et al., 2012). In contrast, parts of the Central Tien Shan and the Eastern Tien Shan receive their largest precipitation input during the summer months. The Pamir mountains receive the highest portion of precipitation during winter and spring months with minimum in summer (Schiemann et al., 2008;Sorg et al., 2012).

Precipitation also exhibits a high spatial variation, ranging from less than 50 mm/year in the desert areas of Tarim and around 100 mm/year on leeward slopes of Central Pamir, to more than 1000 mm/year in the mountain regions, which are exposed to the westerly air flows. These flows are a major moisture source in the region (Aizen et al., 1996;Bothe et al., 2012;Hagg et al., 2013;Schiemann et al., 2008). The combination of the low precipitation in the CA lowlands with high precipitation in the mountains highlight the Tien Shan and Pamir Mountains as the most important regional water source (the so called 'water towers of CA'). Snowmelt in the mountains is the dominant water source for the lowlands during spring and summer, i.e. for most of the vegetation period. During summer, glacier melt and liquid precipitation gain some importance depending on the basin location and degree of glacierisation (Aizen et al., 1996). The Tien Shan and Pamir mountains exhibit particularly high relative water yield compared to the lowland parts of these catchments (Viviroli et al., 2007). Related to the economic water demands in the lowland plains primarily for irrigated agriculture, the Tien Shan and Pamir mountains are among the most important contributors of stream water worldwide (Viviroli et al., 2007). These mountains also have a very high fraction of glacier melt water in summer, particularly in drought years (Pritchard, 2017). Within the Aral Sea basin, to which the

Amudarya and Syrdarya rivers drain, the irrigated area amounts to approximately 8.2-8.4 million ha (Conrad et al., 2016;FAO, 2013). Additionally, considerable irrigation areas are located in the Aksu/Tarim basin, where agricultural land doubled in the period 1989-2011 and land use for cotton production increased even 6-fold (Feike et al., 2015). Irrigated agriculture in Central Asia (CA) is mainly fed by the stream water diversion with only small portion of groundwater withdrawal (FAO, 2013;Siebert

et al., 2010). Hence, reliable prediction of seasonal runoff during vegetation period (April – September) is crucial for agricultural planning and yield estimation in the low lying countries in the Aral Sea basin, as well as for the management of reservoir capacities including dam safety operations in the upper parts of the catchments. Seasonal forecasts are one of the major responsibilities of the hydro-meteorological (hydromet) services of the Central Asian countries and are regularly released starting from January till June with the primary forecast issued end of March – beginning of April for the upcoming

6-months period. In some post-soviet countries, these forecasts are typically developed based on empirical relationships for individual basins relating precipitation, temperature and snow depth/SWE records to seasonal discharge, partly available only in analogue form as look-up tables or graphs (Hydromet Services, unpublished questionnaire survey undertaken within the CAWa project, www.cawa-project.net). Particularly, point measurements of snow depth and/or snow water equivalent (SWE), which have been carried out by helicopter flights or footpath surveys in mountain regions in the past decades, are costly or not

feasible due to access problems nowadays. Other Hydromet Services apply the hydrological forecast model AISHF (Agaltseva et al., 1997) developed at the Uzbek hydro-meteorological Service (Uzhydromet), which computes discharge hydrographs by considering temperature, snow accumulation and melt. Snow pack is accumulated in winter and temperature and precipitation are taken from an analogous year to drive the model in the forecast mode. The hydro-meteorological services rely on meteorological and hydrological data acquired by the network of climate and discharge stations, which, however, strongly

diminished during the 1990s (Unger-Shayesteh et al., 2013). Fortunately the density of the monitoring network recovers nowadays, partly with substantial international support (e.g. Schöne et al. (2013); CAHMP Programme by World Bank; previous programmes by SDC and USAID), but at a slow rate. In any case, the hydro-meteorological services need timely to near real-time data and simple methodologies capable of utilizing available information in order to fulfil their mandatory tasks. Some research activities were undertaken towards the establishment of simple forecast methods in the past, mainly trying to

establish a relationship between large scale precipitation records and seasonal discharge. Schär et al. (2004) showed the potential of the ERA-15 precipitation data from December-April period to explain about 85% of the seasonal runoff variability in May-September in the large-scale Syrdarya river basin. The explained variance for the Amudarya River amounted, however, to only about 25%, presumably due to poor precipitation modelling in the ERA dataset, strong influence of glacier melt and water abstraction for irrigation purposes. Similarly, Barlow and Tippett (2008) explored the predictive power of NCEP-NCAR

cold-season (November-March) precipitation for warm-season (April-August) discharge forecast using canonical correlation analysis. Though for some of the 24 Central Asian gauges, no skillful prediction could be achieved, for a few catchments 20 to 50% explained variance could be attained. Archer and Fowler (2008) utilized temperature and discharge records additionally to precipitation for spring and summer seasonal flow forecast on the southern slopes of Himalaya in northern Pakistan using multiple linear regression models. Despite good predictions of spring and early summer flows, late summer discharges were

poorly forecasted due to the strong influence of summer monsoon. Recently, Dixon and Wilby (2015) demonstrated the skill of a linear regression model for the Naryn basin, Kyrgyzstan, based on TRMM precipitation from October-March to explain 65% of the seasonal flow variance in the vegetation period. The authors selected specific TRMM pixels in the catchments showing the highest correlation to seasonal discharge. They also explored the predictive skill of multiple linear regression models additionally including temperature and antecedent discharge and testing different lead times from one to three months. They showed that forecasts based on multiple linear regression models are always superior to zero order forecasts, i.e. the mean flow.

The fact that substantial snow accumulation in Central Asian mountain regions during the winter and spring months significantly governs runoff in the vegetation period can be effectively utilized for seasonal forecasts. For a similar climatic setting, Pal et al. (2013) included the measurements of snow water equivalent at point locations into multiple linear regression models along with precipitation, antecedent discharge and temperature-based predictors. Linear models with multiple predictor combinations achieved skilful forecasts of the spring (March-June/April-June) seasonal flow in northern India on the southern Himalaya slopes. Point snow measurements are, however, rarely available and remotely sensed snow cover extent can provide a viable alternative. Based on the monitored snow cover extent, e.g. using optical satellite imagery, and additionally considering temperature and precipitation to implicitly approximate snow water equivalent (SWE), a solid basis for seasonal discharge forecast can be assembled. The MODIS snow cover product proved to be highly acccurate for the Central Asian region (Gafurov et al., 2013). Methodologies to remove cloud obstruction of optical imagery have matured over the past decade (Gafurov and Bárdossy, 2009;Gafurov et al., 2016) and tools for automated image acquisition and processing reached the operational level (Gafurov et al., 2016). MODIS snow cover data was e.g. used for runoff forecast in the Argentinian Andes in the high-flow season (September-April), though no cloud elimination algorithms were applied (Delbart et al., 2015). Snow cover in September-October could explain about 60% of the high-flow season discharge variance. However, no skilful forecast with lead times greater than zero were possible. Rosenberg et al. (2011) proposed a hybrid (statistical – hydrological model) framework for seasonal flow prediction in Californian catchments using accumulated precipitation in antecedent period and SWE modelled by a distributed hydrological model. These two predictors were linked to seasonal discharge by principal component and Z-score regression (Rosenberg et al., 2011). The hybrid approach was found comparable and in some cases superior to a purely statistical approach, however, at the cost of substantial efforts for hydrological simulation of the SWE dynamics.

Based on the finding of the studies listed above, we propose a simple methodology for the operational forecast of seasonal runoff for the vegetation period (April-September) for all Central Asian catchments, whose individual drainage areas range over three orders of magnitude. The method is based on multiple linear regression models with automatic predictor selection. The predictors are based on the readily available precipitation, temperature and discharge gauge records, augmented by the operationally processed cloud-free MODIS snow cover product (Gafurov et al., 2016). Based on theoretical considerations it is argued, that in linear modelling the use of meteorological data from a single gauging station for a large catchment is justified, as long as the variability of the station records are representative for the variability within the whole catchment. The validity

of this assumption can be verified by the achieved model performance. We demonstrate the model predictive skill and robustness in a cross-validation and discuss the relative importance of the automatically selected predictors depending on the prediction lead time.

## 2 Study sites and data

For the testing of the forecast models, 13 catchments were selected. The catchments cover a wide range of geographical regions, ranging from catchments along the western slopes of the Altai mountains in Eastern Kazakhstan (Uba, Ulba), the western and northern rim of the Tien-Shan (Chirchik, Talas, Ala-Archa, Chu, Chilik, Charyn) and central Tien-Shan

(Karadarya, Naryn) mountains (cf. Aizen et al., 2007), to the northern and central Pamir (Amudarya) and the northern Hindukush (Murgap). The size of the catchments varies over three orders of magnitude, from 239 $km^2$ to 288,000 $km^2$. Figure 1 provides an overview of the location and size of the catchments, while Table 1 additionally lists the discharge and meteorological gauging stations used for the seasonal flow forecast. Note that the Naryn catchments are nested. The Upper Naryn represents a high alpine catchment and is the headwater catchment of the larger Naryn catchment draining into the

Toktogul reservoir. This separation was undertaken in order to test the proposed method also for a high alpine catchment with a comparably high degree of glacialisation. The wide range of catchment locations, climatic conditions and sizes enable a testing of the proposed forecast models under different boundary conditions, and thus provides an indication of the applicability, robustness and transferability of the approach.

The catchment boundaries are derived to map the catchment area draining to the selected discharge stations. For the

meteorological data (temperature and precipitation) meteorological stations run by the individual Hydromet Services were selected. Ideally those are located in the catchment area and have sufficient data coverage of at least 16 years (starting in 2000 in order to be consistent with the MODIS temporal coverage). However, for two some catchments meteorological stations fulfilling these criteria were not available (Talas, Chilik). For those catchments, meteorological stations nearby were selected for the prediction. For both catchments the meteorological stations are located in the same river catchment downstream of the

discharge station.

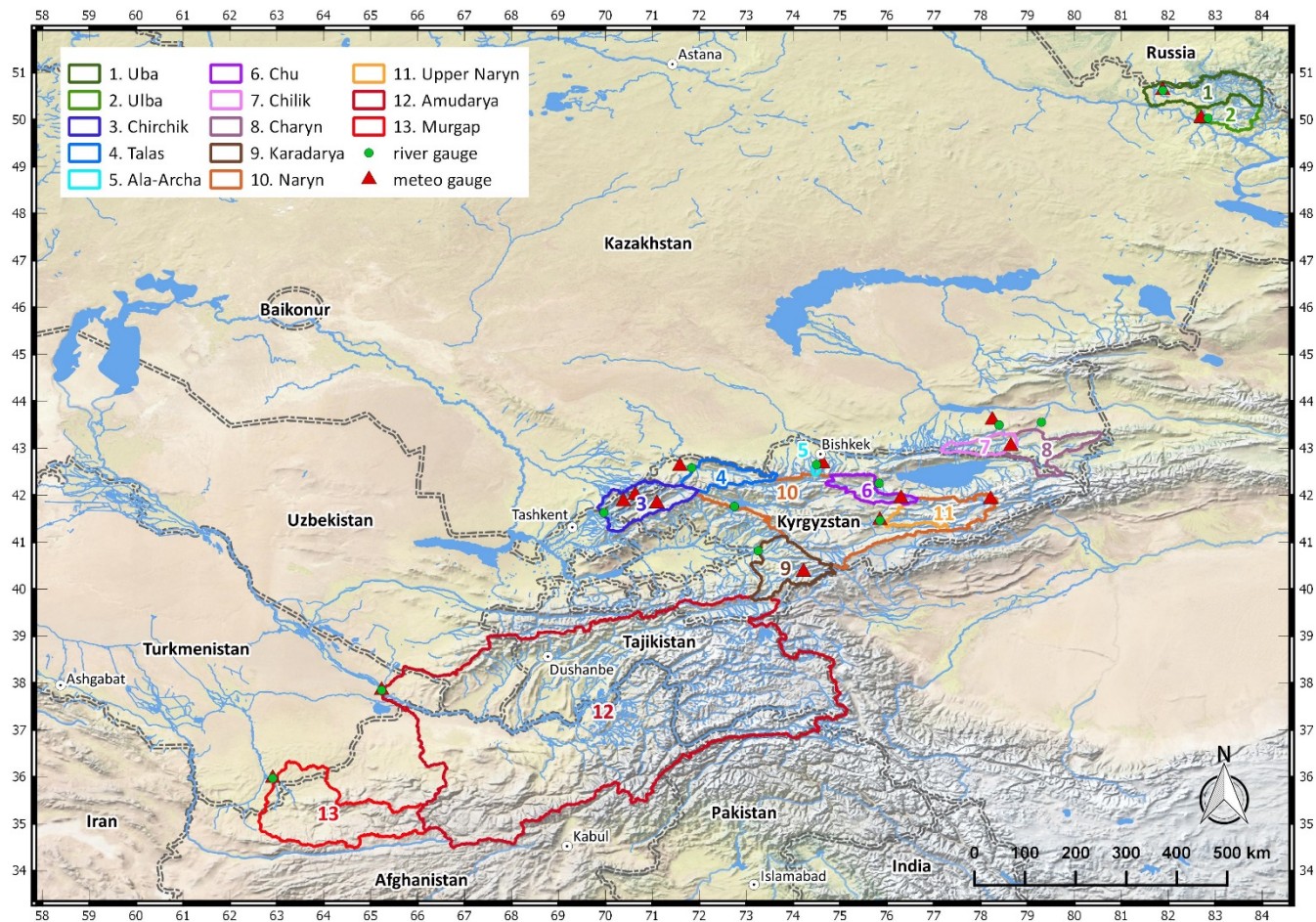

**Figure 1: Overview of the catchments for which prediction models were established, with locations of discharge and meteorological gauging stations used (coordinates in latitude/longitude).**

**Table 1: List of the catchments for which prediction models are derived with discharge (Q) and meteorological gauging stations used for the prediction. Note that Charvak, Andijan and Toktogul are reservoir inflows summing several tributary inflows. For the Charvak reservoir the mean temperature and precipitation data of three meteorological stations located in the catchment was used. Latitude and longitudes are in decimal degrees (WGS84). *Q mean seas.* Is the multiannual mean seasonal discharge from April to September for the period 2000-2015. *Mean ann. P* is the mean total annual precipitation of the meteorological station for the period 2000-2015. *Mean ann. T* is the mean annual mean temperature of the meteorological station for the period 2000-2015. *Mean winter SC* is the mean of the mean daily snow coverage from January to February for the period 2000-2015.**

| | catchment | discharge station | Q deg. lat | Q deg. long | meteo station | meteo deg. lat | meteo deg. long | meteo altitude [m] | catchment area [km²] | Q mean seas. [m³/s] | mean altitude [m] | mean ann. P [mm] | mean ann. T [°C] | mean winter SC [%] |
|---|---|---|---|---|---|---|---|---|---|---|---|---|---|---|
| 1 | Uba | Shemonaikha | 50.620 | 81.880 | Shemonaikha | 50.620 | 81.880 | 300 | 9324 | 269.2 | 740 | 460 | 3.6 | 69.2 |
| 2 | Ulba | Perevalochnaya | 50.033 | 82.843 | Oskemen | 50.030 | 82.700 | 375 | 5080 | 151.4 | 950 | 483 | 3.8 | 87.7 |

| 3 | Chirchik | Charvak | 41.626 | 69.969 | Chatkal | 41.822 | 71.097 | 2300 | 10903 | 346.21 | 2575 | 708 | 5.5 | 97.3 |
|---|---|---|---|---|---|---|---|---|---|---|---|---|---|---|
|  |  |  |  |  | Oygaing | 42.000 | 70.633 | 1620 | 10903 |  |  |  |  |  |
|  |  |  |  |  | Pskem | 41.861 | 70.384 | 2220 | 10903 |  |  |  |  |  |
| 4 | Talas | Kluchevka | 42.581 | 71.836 | Kyzyl-Adyr | 42.616 | 71.586 | 1764 | 6663 | 19.62 | 2424 | 327 | 9.0 | 72.1 |
| 5 | Ala-Archa | Kashka-Suu | 42.650 | 74.500 | Baytik | 42.670 | 74.630 | 1579 | 239 | 8.83 | 3288 | 559 | 3.2 | 79.6 |
| 6 | Chu | Kochkor | 42.250 | 75.833 | Kara Kuzhur | 41.930 | 76.300 | 855 | 4961 | 34.53 | 2934 | 253 | 1.1 | 59.4 |
| 7 | Chilik | Malybai | 43.494 | 78.392 | Shelek | 43.597 | 78.249 | 600 | 3964 | 70.67 | 2603 | 274 | 11.0 | 74.5 |
| 8 | Charyn | Sarytogai | 43.553 | 79.293 | Zhalanash | 43.043 | 78.642 | 1690 | 7921 | 59.06 | 2260 | 507 | 6.1 | 82.4 |
| 9 | Karadarya | Andijan | 40.814 | 73.257 | Ak-Terek | 40.365 | 74.222 | 1190 | 11670 | 186.21 | 2663 | 913 | 9.5 | 82.4 |
| 10 | Naryn | Toktogul | 41.760 | 72.750 | Naryn city | 41.460 | 75.850 | 2040 | 51926 | 653.13 | 2850 | 374 | 4.4 | 88.0 |
| 11 | Upper Naryn | Naryn city | 41.460 | 75.85 | Tien Shan | 41.910 | 78.210 | 3614 | 10343 | 168.64 | 3546 | 345 | -5.8 | 91.0 |
| 12 | Amudarya | Kerki | 37.842 | 65.23 | Kerki | 37.842 | 65.230 | 237 | 287714 | 2551.02 | 2578 | 173 | 17.9 | 56.7 |
| 13 | Murgap | Takhta Bazar | 35.966 | 62.907 | Takhta Bazar | 35.966 | 62.907 | 354 | 35767 | 40.13 | 1707 | 217 | 18.2 | 37.5 |

Monthly values of discharge and meteorological data were obtained for the stations listed in Table 1, i.e. monthly mean discharges, monthly mean temperatures and total monthly precipitation. For the present study meteorological station data was

5 used, because of its operational availability to the CA hydromet Services. Gridded re-analysis products like ERA-Interim typically have a latency of weeks to months, and thus cannot be used for operational forecasts to fulfil the mandatory regulations.  A limitation of station temperature and precipitation data is, that they are likely not representative for basin average values. However, it is assumed that the variability of the catchment averages and the variability of station data is similar. This, in turn, enables the use of the station data in the statistical forecast using multiple linear regressions.

In addition to the station data, mean monthly snow coverages for the individual catchments were calculated using daily snow cover data derived by the MODSNOW-Tool (Gafurov and Bárdossy, 2009;Gafurov et al., 2016). MODSNOW uses the MODIS satellite snow cover product and applies a sophisticated cloud elimination algorithm (Gafurov and Bárdossy, 2009;Gafurov et al., 2016) to obtain cloud free daily snow cover images. The MODSNOW-Tool runs operationally in most of the CA hydromet Services, thus enabling the use of snow cover information for operational forecasts.

Due to the use of MODIS snow cover, which is available since March 2000, the time series of the data used for the construction of the forecast models had to be limited to post-2000. The time period for the model development and testing was thus set to 2000 – 2015, for which mostly complete continuous time series for all data and stations were available.

The seasonal discharge, i.e. the predictand of the forecasts, is calculated as the mean monthly discharge for the period April to September.

## 2.1 Seasonal discharge variability

Figure 2 shows the seasonal discharges for all catchments considered in this study. The top panel highlights the differences in the magnitude of the seasonal discharge, spanning almost three orders of magnitude (cf. also Table 1). Discontinuous lines indicate data gaps. In order to illustrate differences in the inter-annual variability of the seasonal discharge the lower panel of Figure 2 plots the seasonal discharges normalized to zero mean and standard deviation of 1. This plot indicates similar but also different inter-annual variability patterns of the different catchments. In order to distinguish between similar and different inter-annual variabilities cross-correlations of the seasonal discharges are calculated and hierarchically clustered (Figure 3). Cluster memberships were established using the Ward algorithm clustering the catchments based on the dissimilarities of the correlation between the seasonal discharge time series of the different catchments. The correlation matrix in Figure 3 shows that the seasonal discharges mainly cluster according to their geographical location. The variability of the seasonal discharge of the two catchments in the Altai region (Uba, Ulba) is distinctively different to all the others. Also the two most southern catchments (Amudarya and Murgap) form a distinct cluster that is joined by the most western catchment of the northern Tien Shan, Chirchik. However, Chirchik is also well correlated to the largest group, the catchments in the Tien Shan, which all show similar inter-annual variability of the seasonal discharge. An exception to this is the smallest catchment in the study, Ala-Archa, which is not correlated to any of the other catchments, presumably due to the strong influence of local small-scale meteorology and glacier-melt dominated discharge formation in the summer months.

The analysis of the inter-annual variability broadly maps the geographical and climatic differences of the catchments considered in this study. These differences in variability, but also in the magnitude of the discharges and catchment size imply that the forecast methods can be tested against a wide range of boundary conditions and seasonal variabilities. If skilful forecasts are obtained for all catchments, it can be argued that the approach delivers robust forecasts that are not obtained by chance or due to similar variabilities in all catchments. If successful, it could also be inferred that the approach can be transferred to other regions with similar streamflow generation characteristics.

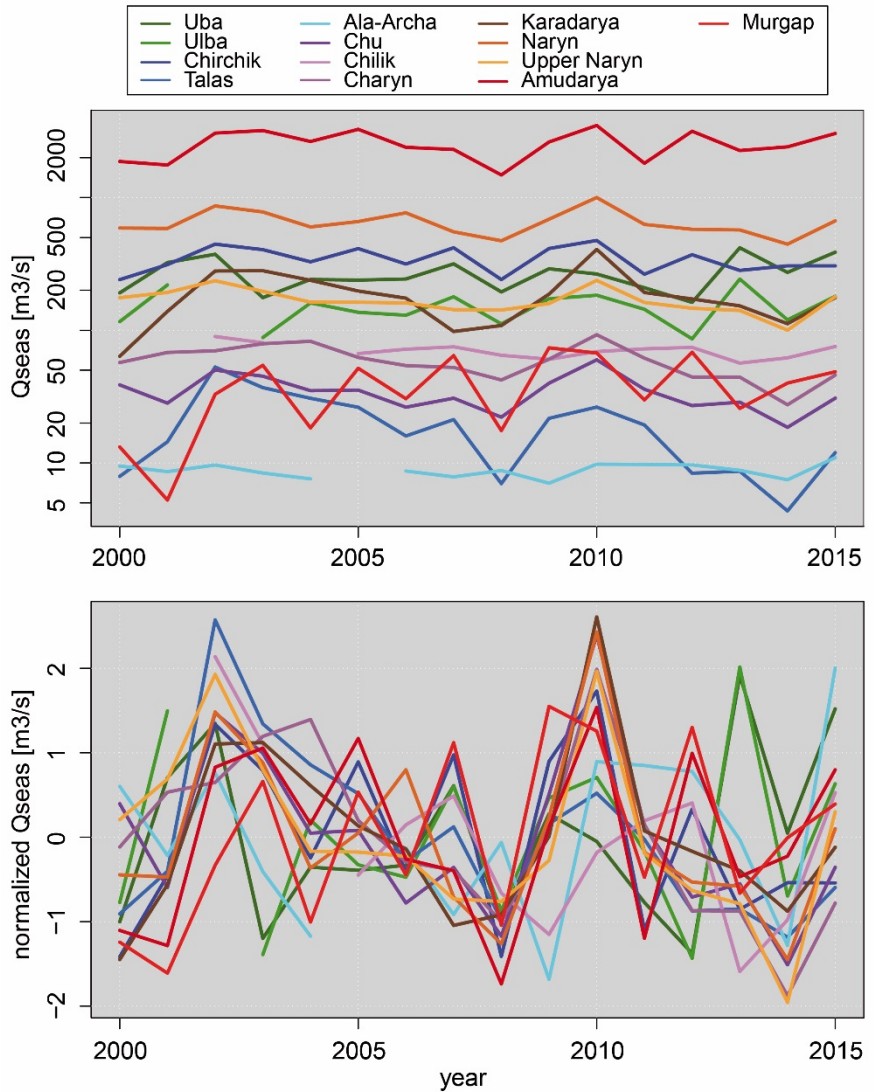

**Figure 2: Seasonal discharge (mean monthly discharge for the period April – September) for the catchments under study (upper panel). The lower panel shows the seasonal discharge normalized to zero mean and standard deviation of 1.**

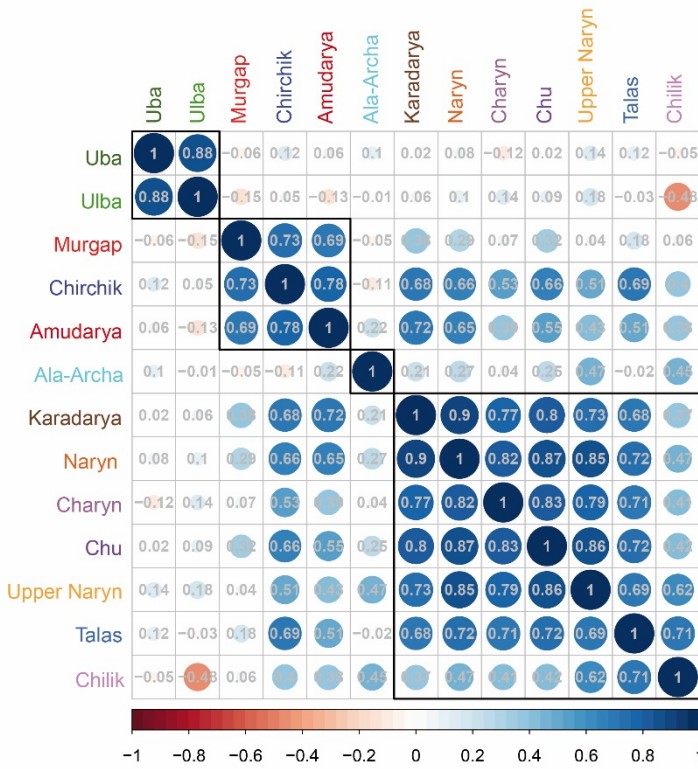

**Figure 3: Correlation matrix of the seasonal discharges of the catchments under study. The catchments are hierarchically clustered using the Ward algorithm. The colour and size of the circles indicate the direction and strength of the correlations, with blue colours indicating positive, and red colours indicating negative correlations. The numbers provide the actual linear correlation coefficient. The coloured circles indicate significant correlation at a significance level of p = 0.05.**

## 3. Method

As mentioned in the introduction, the seasonal discharge during the vegetation period of April to September in CA is dominated by snowmelt in the mountains. Therefore a good estimation of the snow accumulation and snow water equivalent in the catchments during the winter months may provide reliable forecasts of the discharge during the vegetation period. However, snow depth and snow water equivalent are not regularly monitored except for some dedicated research sites. Thus alternative data containing proxy information about the snow depth and water equivalent must be used. Based on these considerations predictors for the forecast models were derived from mean monthly temperature records, total monthly precipitation and monthly mean snow coverage of the catchments. It is argued that combinations of these factors are able to serve as proxy data for snow depth and water equivalent. While the precipitation directly contains information about the snow fall amount and thus accumulation, temperature may contain information on the wetness of the snow pack. In combination with snow coverage,

temperature and precipitation may thus provide information about the snow volume and water content. In addition to the climate data monthly antecedent discharge can serve as an indicator of the magnitude of the snowmelt process and groundwater storage state and release, and is used as predictor, too. The usefulness of antecedent discharge for flow prediction has been shown e.g. in Slater et al. (2017). It has to be noted that in this study only direct observations are used. Forecasted climate data are not used for the predictions.

For some regions, particular the Altai catchments, early summer (May – July) precipitation plays a larger role for the seasonal discharge. This precipitation is partly considered as observations in the late forecasts presented here. However, reliable information about the May-July precipitation already in spring could possibly improve the early forecasts. But due to the low predictability of the typically convection type summer precipitation (Gerlitz et al., 2016), this is not considered in the predictor set.

Evaporative losses in the presented mountain catchments are considered low due to the low summer temperatures, fast catchment response, and high water flow velocities in the rivers. Higher losses can occur in reservoir lakes, but with the exception of the large Amudarya basin no reservoirs are present in the selected catchments. In the Amudarya catchment the Nurek reservoir lake at the Vakhsh river exist, but evaporative losses from the lake surface area of 98 km$^2$ can be considered negligible in comparison to the large catchment size. Therefore evaporation is not directly considered as predictor for the forecasts.

The catchment area of the Vaksh river at the conjunction with the Panj river amounts to 31,415 km², equivalent to about 11% of the Amudarya catchment at Kerky considered here. Assuming further that the reservoir can manage only a fraction of the total discharge of the Vakhsh river, and that the effects of the water retention are further buffered by the seasonal mean discharge spanning six months, it can be assumed that the regulating effect of the Nurek dam on the overall seasonal discharge of the Amudarya at Kerky is rather low. Additionally, the dam is operational since 1980, therefore a discontinuity in the time series 2000-2015 can be ruled out. We thus argue that the anthropogenic influence of the seasonal discharge time series of the Amudarya is negligible for the presented study.

### 3.1 Generation of the predictor set

The core set of predictors consists of the monthly values preceding the prediction date. According to the operational forecast schemes of the CA Hydromet Services a series of different prediction dates were defined. The first prediction of the seasonal mean discharge for the vegetation period (April to September) is issued on January 1$^{st}$, followed by predictions on February 1$^{st}$, March 1$^{st}$, April 1$^{st}$, May 1$^{st}$, and June 1$^{st}$. The predictions January to March are preliminary forecasts, while the prediction on April 1$^{st}$ is the most important for the water resource planning in the CA states. The following operational forecasts serve as corrections of the April forecast. However, if the discharge of the whole vegetation season is predicted in these late forecasts, the predictors for the late forecasts are not fully independent from the predictand, because the discharge of April and May is included in the seasonal discharge, but at the same time used as predictor. This violates formal requirements of a linear regression and biases the performance evaluation of the late predictions. Therefor we deviate from the official procedures of

the CA Hydromet Services for the late forecasts. In this study the mean discharge of the remaining vegetation season is predicted in the late forecasts (i.e. May-September for the May forecast, and June-September for the June forecast), and the observed discharge in April and May is then added to the forecasts in order to obtain values for the whole vegetation period. By this procedure the formal requirements of the MLR are fulfilled, and the results of the forecasts are in line with the formal procedures of the hydromet services. This is necessary to obtain acceptance of the proposed method in the services and their use in the official forecast procedures, because the water regulation procedures and e.g. agricultural yield estimations are traditionally based on bulk numbers for the entire period. If these procedures are not followed, the obtained results, which are better than the forecasts issued with the existing procedures, might not be implemented and come into practise, and thus a chance would be missed to bring research results into application. In this context it has to be noted that sub-seasonal discharges are highly correlated to the full seasonal discharge (cf. Annex 1). This means in consequence that regression results obtained with sub-seasonal discharges will be very similar to results obtained using the full discharge time series.

The selection of the predictor set to be used in the MRL follows this rational, which is based on a combination of principle hydrological considerations, preliminary correlation tests and expert judgement: For the prediction up to the 1$^{st}$ of April the monthly values over the whole winter period, i.e. from October to the prediction date are used. For later predictions (i.e. in May and June) the set was limited to data of the prediction year, i.e. from January onwards. This restriction was implemented in order to keep the number of predictor combinations and thus the calculation time in reasonable limits. It can be justified by the typically low correlation of the October to December data with the seasonal discharge compared to later values. The monthly predictor values were accompanied by multi-monthly means, spanning over two and three months prior to the prediction date, and mean values for the whole predictor period defined above, i.e. either from October to the prediction month, or from January to the prediction month, respectively. An exception for this is the snow coverage, which is typically invariant in the deep winter months (100% coverage). Therefore multi-monthly means are only calculated for the whole predictor period for the early forecasts in January and February. For later forecast multi-monthly means from January onwards are used. The early winter predictors are not used in the March to June forecasts because of their low predictive power, which is illustrated by the comparatively low performance of the early forecasts (cf. section 4).

Furthermore, composites were calculated from the climatological data in order to extend the predictor set. They are introduced in order to explore their potential to reflect snow wetness. It is argued that composites can improve the prediction by linear models, as some non-linear interactions might be reflected better by composites compared to the raw data (as shown in e.g. Hall et al., 2017). Analogously to the original data, monthly and multi-monthly composites were derived. For the composites, products of "temperature and precipitation", "temperature and snow coverage", "precipitation, snow coverage and temperature", "precipitation and snow coverage" were used. Antecedent discharge was not included in the composites, because this should not influence the snow cover characteristic.

## 3.2 Statistical modelling

For the development of the statistical forecast models standard multiple linear regression (MLR) was applied. It is argued that the discharge generation from snowmelt over whole catchments and on a seasonal time scale can be approximated by linear models. In fact, this was shown by a large number of studies using hydrological models based on linear concepts like linear storage, e.g. Duethmann et al. (2014) and Duethmann et al. (2015) in Central Asia. Additionally a number of studies have shown that linear regression is a valid approach for seasonal forecasts (e.g. Delbart et al., 2015;Dixon and Wilby, 2015;Seibert et al., 2017). A linear modelling approach is thus seen as a valid approach for seasonal forecasts in the study region from a general point of view. However, in order to statistically support the assumption that the runoff generating processes can be approximated by linear models, the formal assumptions of MLR were also tested: the assumption of normal distribution of the residuals was tested by the Shapiro-Wilk test, the independence of the residuals was tested by calculating the autocorrelation with lag 1, and the heteroscedasticity of the residuals was tested by the Breusch-Pagan test.

In the model selection procedure all possible predictor combinations, which are different for every prediction month as described in 3.1, are used in the MLR for the construction of forecast models. However, some restrictions were put on the predictor combinations in order to avoid overfitting and thus spurious regression results:

1. The predictors are grouped into 8 groups: snow cover, temperature, precipitation, antecedent discharge, and the four composite types.

2. The maximum number of predictors in a regression is limited to four.

3. Only one predictor from each group of predictors can be used in an individual regression model.

This resulted in 7,728 predictor combinations, i.e. multiple linear models to be tested in January, and increased to 155,690 possible models in April. A complete list of the predictors for the different prediction months is provided in Annex 2. The coefficients for all these linear models were automatically fitted during the MLR by the least squares method. Only models with all predictors statistically significant at $p = 0.1$ and with an overall model significance of at least $p = 0.1$ were retained. From the remaining models the best models were selected based on the lowest Predicted Residual Error Mean of Squares (PREMS) value obtained by a Leave-One-Out Cross Validation (LOOCV). In the LOOCV one year of the seasonal discharge time series is removed from the data set for fitting the MLR. The missing data point is then estimated by the model fitted to the remaining data. The PREMS value is the mean of squared errors of all seasonal discharges left out and the associated predicted LOOCV values. PREMS is thus defined as:

$$PREMS = \frac{1}{n}\sum_{i=1}^{n} e_{(i)}^{2}$$

with $e_{(i)}$ being the residuals of the LOOCV:

$$e_{(i)} = |y_i - \hat{y}_{(i)}|$$

where $\hat{y}_{(i)}$ is the regression estimate of $y_i$ based on a regression equation computed leaving out the i[th] observation of the overall number of $n$ observations. The PREMS was used in this study instead of the usual PRESS (Predictive Residual Error Sum of Squares) in order to avoid biases possibly introduced by missing predictor or predictand data. Using the sum of squares could favour models with missing data compared to models providing predictions for all 16 years. Using the mean of the squares can avoid this to a large extent.

The LOOCV is testing the MLR for robustness and can avoid overfitting and incidental good MLR results valid for the whole data set only. In order to avoid an over-estimation of the forecast skill the seasonal discharge time series were tested for auto-correlation, which could lead to spurious estimation of model robustness in the LOOCV.

Model skill was evaluated by a number of measures: adjusted $R^2$, root mean square error RMSE, and mean absolute error MAE. The robustness of the model set was quantified as the ratio between the adjusted $R^2$ based on the LOOCV residuals and the adjusted $R^2$ of the complete model residuals. The reliability of the model was analysed by PIT diagrams (e.g. Crochemore et al., 2017) and quantified as PIT-scores (Renard et al., 2010).

In the presented study not only the single-best model according to PREMS of the LOOCV-MLR was selected as prediction model, but rather the best 20 models, if more than 20 models pass the significance tests. This selection aims at the analysis of the differences between the best models in terms of performance and predictors, but also serves as a set of models for the forecast of the seasonal discharge. The distribution of the residuals of the best 20 forecast models was evaluated to provide 80% predictive uncertainty bounds for every forecast. However, it has to be noted that the choice to use the best 20 models is subjective, and this number can be increased or reduced. Moreover, a distinct set of specific models from the best models can be selected according to their performance measures and temporal dynamics by experts knowledgeable of the individual catchments. Sufficient amount of freedom was left for the selection of the number and individual selection of best models to be used for the forecasts, in order to enable an expert selection of models by the forecasters of the Central Asian Hydromet Services. The forecasters can check every model retained for their performances (quantitatively and qualitatively), and select the models to be used for the prediction accordingly.

## 3.3 Predictor importance

The predictors of the selected best models were analysed for their importance, i.e. their share of the overall explained variance ($R^2$) of the individual models. This was achieved by the *lmg* algorithm implemented in the R-package *relaimpo* (Grömping, 2006). *lmg* is based on sequential $R^2$s, but explicitly eliminating the dependence on predictor orderings by averaging over orderings using simple unweighted averages. In sequential $R^2$ calculations, the model is re-run with a single predictor only and the explained variance is calculated. Then the next predictor is added and the gain in explained variance is calculated. By this procedure the variance explained by individual predictors can be quantified. However, in this procedure the sequence of predictors added influences the share of explained variance associated to the individual predictors. Therefore the *lmg* algorithm tests all possible predictor sequences and calculates the mean importance of every sequence in order to overcome the problem

of predictor ordering in sequential $R^2$s. The predictor importance calculation yields information about the importance of the individual predictors at different forecast points in time for the catchments under study. This can be used for a discussion of the factors responsible for the winter snow accumulation and snow water content in the catchments.

However, such a discussion is complicated by the use of the composite predictors. Therefore the importance of composite predictors is divided into equal proportions to the components of the composites. If more than one composite is used in a model, the proportions associated to the component factors (snow cover, precipitation, temperature) are summed up and displayed as parts of the composite importance in the figures presented in 4.2. This analysis is not meant to provide a quantitative estimation for the component importance of the composite predictors, but rather to enhance the discussion and interpretation of the predictors of the selected forecast models.

In addition to the importance for an individual model (here the best LOOCV model), the mean importance over the best 20 LOOCV models is calculated. This is achieved by calculating the fractions of the sum of importance of an individual predictor for all 20 models to the sum of the $R^2$ values of all 20 models for each catchment and month. These fractions are then multiplied by the mean $R^2$ values of the best 20 models. This mean predictor importance can be compared to the predictor importance of the best model in order to analyse the stability of the predictor selection within the best 20 LOOCV models.

## 4. Results

In order to test the suitability of the LOOCV for the seasonal streamflow forecast the autocorrelation and partial autocorrelation of the streamflow time series was calculated and plotted (Annex 2). Any autocorrelation in the discharge time series could lead to artificial over-estimation of the forecasts skill by the LOOCV. Hardly any autocorrelation at $\alpha = 0.05$ could be detected. Only for the Ulba some significant autocorrelation for lag 1 and 2 is shown just above $\alpha = 0.05$, but by the partial autocorrelation only. No autocorrelation was found at $\alpha = 0.01$. Therefore it can be stated that autocorrelation does not exist in the discharge time series of all catchments, and thus the proposed LOOCV is an appropriate validation method.

The MLR fitting with LOOCV (cf. 3.2) was applied for different forecast dates ranging from January 1st to June 1st for all catchments. Out of the models that passed the significance tests after fitting the best 20 models according to the PREMS resulting from the LOOCV were retained for the forecasts. The tests for possible violations of the formal MLR assumptions showed, that 89.5% of all selected models for all catchments and prediction months fulfilled the criteria of normal distributed residuals, 95.8% of the selected models passed the test for independence of the residuals, and 99.5% of the selected models have homoscedastic residuals (cf. Annex 3). In summary, the formal requirements of MLRs are fulfilled by almost all models, and the use of linear models for seasonal discharge forecast is justified also from a formal point of view.

In general the performance of the linear models increases from January to June, with the best models reaching adjusted $R^2$ (adj. $R^2$) values in the range of $0.68 - 0.97$ in April and $0.89 - 0.99$ in June. For most of the catchments high adj. $R^2$ values in the range of $0.57 - 0.83$ were already obtained in January. Only for Ala-Archa, Amudarya, Chu and Chirchik the performance

is unsatisfyingly low in January, but increases to adj. $R^2 > 0.59$ already one month later in February. Table 2 lists the adj. $R^2$ values of the best LOOCV models for all catchments and forecast months. Note that the adj. $R^2$ in the table are calculated using the coefficients of the linear models fitted to the whole data set, i.e. they are not cross validated adj. $R^2$. For the months May and June the performance is a) calculated for the whole seasonal discharge by adding the forecast values of the sub-season

to the observed discharge, and b) for the sub-season only. As expected, the adj. $R^2$ values are higher for the whole seasonal discharge, because in this case parts of the seasonal discharge is explained by actual observations. However, the performance is just slightly lower for the sub-seasonal discharge predictions, typically in the range of $0.01 - 0.05$ less of the adj. $R^2$ for the wholes season. Only for Uba a larger difference of about 0.1 in adj. $R^2$ in May is observed. This good performance of the sub-seasonal forecasts can be explained by the general high correlation of the sub-seasonal discharges with the full seasonal

discharge (Annex 1), and the high predictive power of antecedent discharge for the later forecasts (cf. section 4.2).

While for most of the catchments, the performance of the models gradually increases with decreasing lead time including the sub-seasonal discharge forecasts, the performance for Chilik, shows significant decreases and increases. This is mainly caused by a comparatively large number of missing discharge and predictor data, but possibly also by the fact that the meteorological station used for this catchment is located outside of the catchment. The automatic fitting algorithm takes advantage of this by

finding models able to explain the fewer data points better compared to the full time series despite the use of PREMS instead of PRESS. However, these models can already represent an overfitting and are thus less reliable or stable in time compared to models fitted to longer time periods.

In order to get a more comprehensive picture of the model performance, Figure 4 shows the temporal evolution of the adj. $R^2$ evaluated for the complete time period of the single best LOOCV model, the minimum adj. $R^2$ of the best 20 models, the mean

adj. $R^2$ of the best 20 models, the root mean square error (RMSE) of the single best LOOCV model calculated for the full data set normalized to the mean seasonal discharge (cf. Table 1), the normalized mean absolute error MAE, and the PREMS value of the best model. Note that the highest adj. $R^2$ value is not necessarily the adj. $R^2$ of the single best model, because the best model is selected according to the lowest PREMS in the LOOCV, and not the best adj. $R^2$ evaluated using the whole time series. Therefore the mean adj. $R^2$ in January is occasionally higher than the adj. $R^2$ of the best LOOCV model, i.e. the most

robust model. In general, Figure 4 shows that the different adj. $R^2$, RMSE, MAE and PREMS values are similar in their evolution in time, i.e. increase (adj. $R^2$), resp. decrease (RMSE, MAE, PREMS) with later forecast months. This indicates that for all best 20 models the performance is improving with later forecasts.

Furthermore, the difference between min adj. $R^2$ and mean adj. $R^2$ to the adj. $R^2$ of the single best LOOCV model is typically larger in the early prediction months. This indicates a wider spread of model performance within the selected 20 models for

the predictions with longer lead times. This difference decreases with shorter lead times, meaning that more models with similar high performance can be found, and thus uncertainty of the model set is reduced. To a certain extent this is likely caused by the larger number of possible predictors for later prediction months, but it is also well justified to assume that the later predictors have more predictive power: data from the late winter months can better describe the snow coverage and water content compared to predictors from the previous autumn/early winter. This issue will be discussed further in Section 4.3.

Figure 4 shows that the RMSE as well as the MAE of the best model of the LOOCV is at maximum about 35% of the long term seasonal mean discharge (Talas and Murgap in January). However, for most catchments the normalized RMSE and MAE is below 20% in January already. For the important April forecast they fall generally below 10%, except for Talas and Murgap, where it remains at 20%. These values state the high performance of the linear forecast models in terms of actual discharge,

and are thus a useful information for practitioners in order to assess the value of the forecasts.

Figure 4 also shows the PREMS values of the best models and the performance development with the forecast months. The PREMS values generally decrease (i.e. improve) with decreasing lead time. However, occasionally increases can be observed for later forecast months. This can be also seen in the adj. $R^2$ values, but less pronounced because of the scale of the left y-axis. This phenomenon is caused by the changing predictor sets from forecast month to forecast month. Particularly multi-

monthly predictors change for each prediction date according to the parameter selection outlined in Section 3.1. As this phenomenon of increasing PREMS values usually occurs in April or May, it can be hypothesized that the information of the late winter/early spring months used in the later forecasts does not contain better information about the snow cover as the previous months. With respect to a practical application, the better performing forecasts from the previous months can be used, which is equivalent to an extension of the predictor set by the predictors of the previous month.

This general reduction of PREMS also means that the models become more robust for later prediction months. To illustrate this more clearly, Figure 4 also shows the relation between the mean adj. $R^2$ of the LOOCV for all 20 models to the mean adj. $R^2$ of the full model fit. The mean adj. $R^2$ of the LOOCV is calculated from the LOOCV residuals used to calculate the PREMS. According to the rationale of the LOOCV, a model is more robust and less prone to overfitting, if the LOOCV-$R^2$ is very close to the overall $R^2$. Figure 4 shows that this is generally the case for the catchments with very high adj. $R^2$ values, and also for

later prediction months. This means that the selection of the predictors is likely stable even if additional data is added to the time series in future. However, there are some catchments for which comparably less robust models could be derived even for later prediction months (5. Ala-Archa, 6. Chu). For these catchments it is likely that the predictor selection will change with additional data.

**Table 2: Adjusted $R^2$-values of the best performing prediction models from the LOOCV for all catchments and prediction months. "best" indicates the single best model according to the LOOCV, "mean" indicates the mean percentage over the best 20 models according to the LOOCV. The adjusted $R^2$ values are associated with indicators for significance levels. Additionally, for the prediction months May and June the adjusted $R^2$ values for the prediction of the remaining season (i.e. for the mean seasonal discharge May-September for prediction month May, and June-September for prediction month June) are given in italics.**

| | | January | | February | | March | | April | | May | | June | |
|---|---|---|---|---|---|---|---|---|---|---|---|---|---|
| | | best | *mean* | best | *mean* | best | *mean* | best | *mean* | best | *mean* | best | *mean* |
| 1 | Uba | 0.678 ++ | 0.511 ++ | 0.824 +++ | 0.714 +++ | 0.842 +++ | 0.743 +++ | 0.811 +++ | 0.790 +++ | 0.868 *0.775* +++ | 0.828 *0.723* +++ | 0.968 *0.836* +++ | 0.958 *0.793* +++ |
| 2 | Ulba | 0.624 o | 0.429 + | 0.714 +++ | 0.444 + | 0.781 +++ | 0.672 ++ | 0.869 +++ | 0.811 +++ | 0.886 *0.858* +++ | 0.919 *0.875* +++ | 0.987 *0.921* +++ | 0.979 *0.862* +++ |

| # | Name | | | | | | | | | | | | |
|---|---|---|---|---|---|---|---|---|---|---|---|---|---|
| 3 | **Chirchik** | 0.253 ++ | 0.278 -- | 0.594 +++ | 0.556 ++ | 0.650 +++ | 0.593 ++ | 0.891 +++ | 0.884 +++ | 0.959 *0.931* +++ | 0.940 *0.918* +++ | 0.972 *0.948* +++ | 0.965 *0.932* +++ |
| 4 | **Talas** | 0.669 +++ | 0.408 + | 0.794 +++ | 0.703 +++ | 0.808 +++ | 0.728 +++ | 0.823 +++ | 0.787 +++ | 0.921 *0.894* +++ | 0.872 *0.831* +++ | 0.966 *0.940* +++ | 0.958 *0.925* +++ |
| 5 | **Ala-Archa** | 0.393 + | 0.353 o | 0.597 ++ | 0.431 o | 0.758 +++ | 0.524 + | 0.761 +++ | 0.623 ++ | 0.742 *0.733* ++ | 0.635 *0.603* ++ | 0.820 *0.772* +++ | 0.778 *0.719* ++ |
| 6 | **Chu** | 0.274 + | 0.260 -- | 0.709 +++ | 0.440 o | 0.903 +++ | 0.729 +++ | 0.680 +++ | 0.569 ++ | 0.812 *0.805* +++ | 0.754 *0.745* +++ | 0.895 *0.882* +++ | 0.858 *0.841* +++ |
| 7 | **Chilik*** | 0.865 +++ | 0.818 ++ | 0.856 +++ | 0.787 ++ | 0.910 +++ | 0.873 +++ | 0.757 +++ | 0.770 ++ | 0.884 *0.843* +++ | 0.800 *0.770* +++ | 0.934 *0.934* +++ | 0.829 *0.831* +++ |
| 8 | **Charyn** | 0.643 +++ | 0.503 + | 0.844 +++ | 0.786 +++ | 0.792 +++ | 0.765 +++ | 0.873 +++ | 0.810 +++ | 0.943 *0.918* +++ | 0.947 *0.919* +++ | 0.989 *0.964* +++ | 0.986 *0.951* +++ |
| 9 | **Karadarya** | 0.573 ++ | 0.449 + | 0.589 +++ | 0.411 ++ | 0.880 +++ | 0.845 +++ | 0.976 +++ | 0.968 +++ | 0.994 *0.988* +++ | 0.984 *0.970* +++ | 0.983 *0.947* +++ | 0.978 *0.930* +++ |
| 10 | **Naryn** | 0.782 +++ | 0.679 +++ | 0.657 +++ | 0.657 +++ | 0.844 +++ | 0.800 +++ | 0.853 +++ | 0.819 +++ | 0.824 *0.811* +++ | 0.824 *0.806* +++ | 0.932 *0.896* +++ | 0.898 *0.847* +++ |
| 11 | **Upper Naryn** | 0.832 +++ | 0.810 +++ | 0.898 +++ | 0.850 +++ | 0.916 +++ | 0.897 +++ | 0.947 +++ | 0.923 +++ | 0.862 *0.847* +++ | 0.863 *0.839* +++ | 0.946 *0.917* +++ | 0.944 *0.916* +++ |
| 12 | **Amudarya** | 0.213 + | 0.304 + | 0.841 +++ | 0.691 +++ | 0.857 +++ | 0.840 +++ | 0.878 +++ | 0.839 +++ | 0.897 *0.869* +++ | 0.893 *0.845* +++ | 0.982 *0.961* +++ | 0.977 *0.949* +++ |
| 13 | **Murgap** | 0.465 ++ | 0.367 o | 0.757 +++ | 0.551 + | 0.802 +++ | 0.642 ++ | 0.807 +++ | 0.700 ++ | 0.985 *0.965* +++ | 0.970 *0.934* +++ | 0.997 *0.985* +++ | 0.996 *0.981* +++ |

\* the performance of Chilik is not representative and comparable to the other catchments due to too many missing discharge and predictor data.

Significance level p: +++ = 0.01, ++ = 0.05, + = 0.1, o = 0.2, -- = >0.2; for mean the lowest significance of the model set is used.

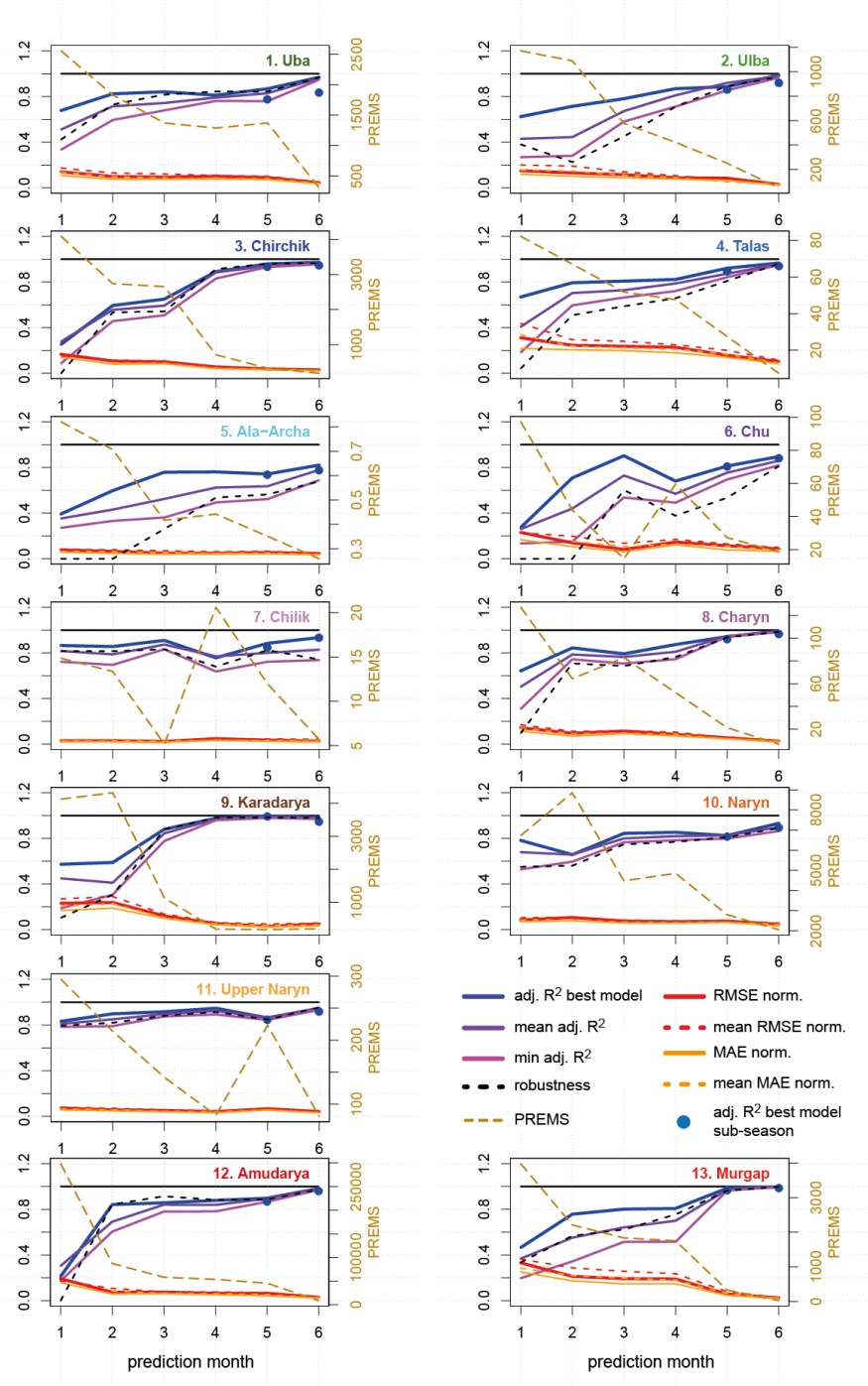

**Figure 4: Performance of the prediction models for the different catchments and prediction months.** *Adj. $R^2$ best model* is the adjusted $R^2$ of the single best LOOCV model given for the prediction of the complete seasonal discharge (lines) and the sub-seasonal discharge prediction in May and June (dots); *mean adj. $R^2$* is the mean adj. $R^2$ of the best 20 LOOCV models; *min adj. $R^2$* is minimum adj. $R^2$

of the best 20 LOOCV models; *robustness* is mean LOOCV-adj. $R^2$ of the best 20 models divided by the mean adj. $R^2$; *RMSE/MAE norm.* is the root mean squared error/mean absolute error of the single best model normalized to mean multi-annual seasonal discharge; *mean RMSE/MAE norm* is the mean root mean square error/mean absolute error of the best 20 LOOCV models normalized to the multi-annual mean seasonal discharge; *PREMS* is the predictive residual sum of squares (PRESS) of the single best model, divided by the number of years for which prediction could be made (i.e. predictive residual mean of squares). Prediction months 1-6 refer to predictions made in January to June.

In addition to the performance metrics Figure 5 plots the temporal dynamics of the best LOOCV models for all six prediction months. It can be seen that the models can map the high variability of the observed seasonal discharges very well, often already in January or February. This graphically corroborates the findings derived from the performance metrics and underlines that the good performance of the models is not a statistical artefact.

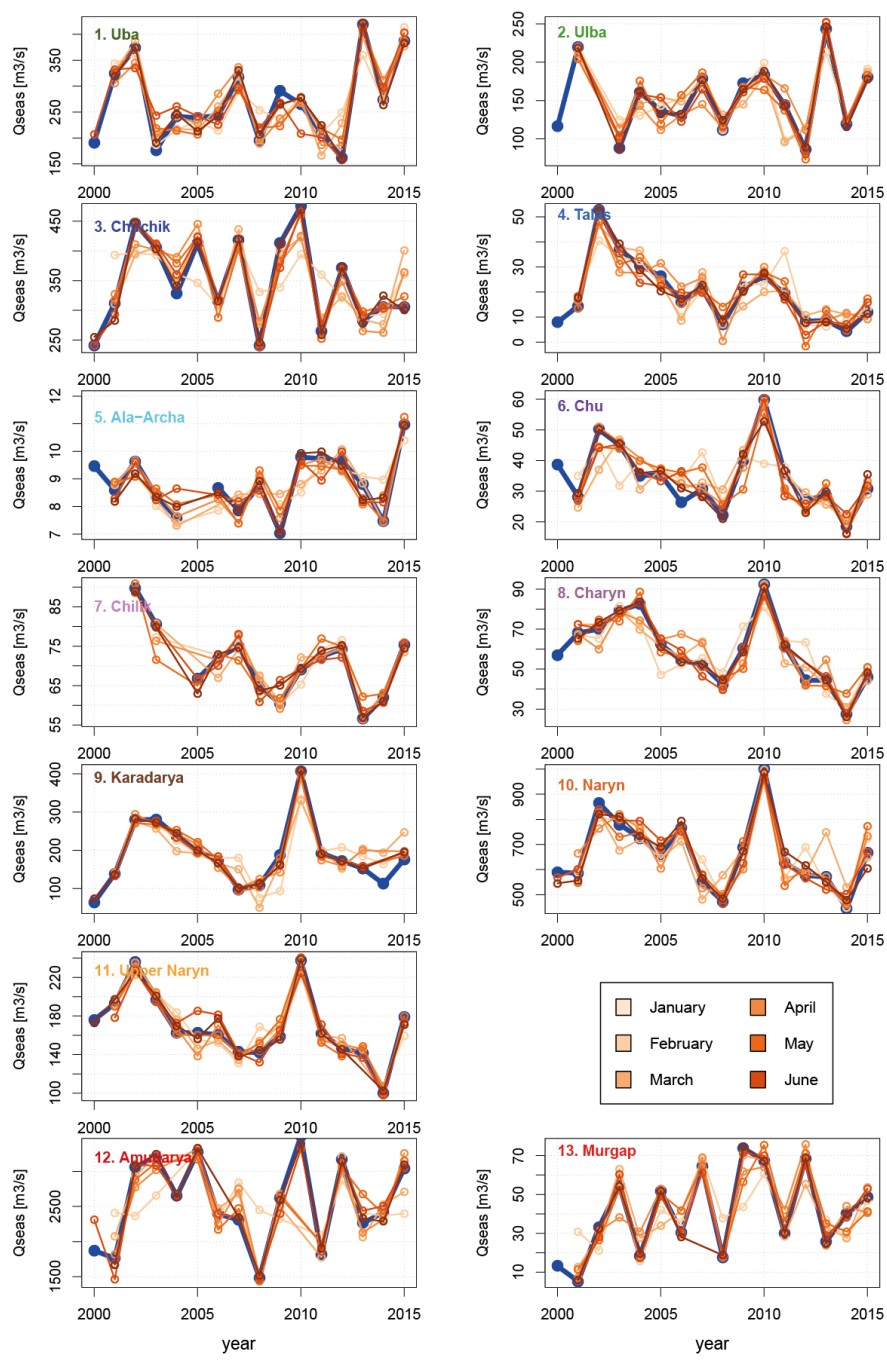

**Figure 5: Forecasts of the seasonal discharge by the single best model selected by the LOOCV for the individual catchments and all prediction months. The blue lines show the observed seasonal discharges. Note that some models do not provide forecasts for every year due to missing predictor data.**

In order to set the performance of the presented models in the context of the routines and guidelines of the Central Asian Hydromet Services, the performance of the models was also estimated according to the performance criteria used by the Hydromet Services. This is defined by:

$$S_\sigma = \frac{|res|}{\sigma_{Qs}}$$ (1)

With |res| denoting the absolute value of the residual of an individual forecast, and $\sigma_{Qs}$ the standard deviation of the seasonal discharge (here calculated for the discharge time series used, i.e. for the period 2000-2015). According to the protocols of the Hydromet Services an acceptable ("good") forecast is defined by $S_\sigma < 0.675$. Table 3 shows how often this criteria was fulfilled during the analysis period 2000-2015 for the best model, and on average by the best 20 models. For the critical forecast month April the criteria was fulfilled for at least 81% of the years (13 out of 16 years) by the best model for all catchments. For all catchments the percentages increase further for the later forecast months. These findings are also valid for the set of the best 20 models, as the very similar percentages of the mean of all models compared to the best model indicate. This means that the developed models would provide acceptable forecasts for the Hydromet Services in the range of 80%-90% for the important forecast month April.

**Table 3: Number of times the models yield acceptable prediction according to the criteria of the Central Asian Hydromet Services for all catchments and prediction months. Numbers indicate percentage of the years of the period 2000-2015 for which the criteria for an acceptable forecast is fulfilled. "best" indicates the best model according to the LOOCV, "mean" indicates the mean percentage over the best 20 models according to the LOOCV.**

| | | January | | February | | March | | April | | May | | June | |
|---|---|---|---|---|---|---|---|---|---|---|---|---|---|
| | | best | *mean* | best | *mean* | best | *mean* | best | *mean* | best | *mean* | best | *mean* |
| 1 | Uba | 69% | 70% | 88% | 85% | 88% | 85% | 88% | 83% | 93% | 93% | 100% | 100% |
| 2 | Ulba | 80% | 62% | 87% | 71% | 87% | 77% | 93% | 87% | 100% | 99% | 100% | 100% |
| 3 | Chirchik | 50% | 54% | 75% | 73% | 75% | 75% | 88% | 93% | 100% | 100% | 100% | 100% |
| 4 | Talas | 81% | 67% | 94% | 82% | 88% | 81% | 88% | 88% | 100% | 99% | 100% | 100% |
| 5 | Ala-Archa | 67% | 59% | 73% | 63% | 80% | 69% | 87% | 75% | 85% | 81% | 92% | 87% |
| 6 | Chu | 69% | 55% | 81% | 70% | 88% | 81% | 81% | 77% | 93% | 90% | 100% | 96% |
| 7 | Chilik | 85% | 83% | 85% | 82% | 85% | 93% | 92% | 87% | 92% | 92% | 100% | 93% |
| 8 | Charyn | 75% | 67% | 88% | 84% | 88% | 83% | 94% | 88% | 100% | 100% | 100% | 100% |
| 9 | Karadarya | 75% | 70% | 69% | 69% | 88% | 84% | 88% | 88% | 100% | 100% | 100% | 100% |
| 10 | Naryn | 88% | 79% | 75% | 79% | 88% | 84% | 88% | 87% | 100% | 99% | 100% | 100% |
| 11 | Upper Naryn | 88% | 86% | 88% | 87% | 88% | 90% | 94% | 92% | 100% | 96% | 100% | 100% |
| 12 | Amudarya | 44% | 51% | 81% | 70% | 75% | 79% | 81% | 82% | 94% | 96% | 100% | 100% |
| 13 | Murgap | 75% | 66% | 88% | 76% | 88% | 78% | 88% | 86% | 100% | 100% | 100% | 100% |

## 4.1 Predictive uncertainty

In order to quantify the predictive uncertainty the empirical 10% and 90% percentiles of the residuals of the forecasts sets
consisting of the up to best 20 models according to PREMS were calculated for every prediction month. Note that for the early
prediction months occasionally less than 20 models passed the significance test. The tables in Annex 3 indicate when this was
the case. The quantiles of the residuals were then added to the median of the predictions of the model set, thus providing an
80% predictive uncertainty band, i.e. an interval in which the true value of the seasonal discharge should lie with a probability
of at least 80%. Figure 6 shows the predictive uncertainty bands for every catchment along with the observed seasonal
discharge. The predictive uncertainty for the different prediction months are shown in shades of orange. In general it can be
seen, that the predictive uncertainty bands narrow with later prediction months, illustrating the better prediction during later
prediction months described above. While this is perfectly visible for most catchments (e.g. 3. Chirchik, 7. Karadarya), it is
not the case for some others (5. Ala-Archa, 6. Chu, 7. Chilik, 10. Naryn). For Chilik and Naryn this is a consequence of the
already high performance of the early forecasts, which results in similar uncertainty bands for the different prediction months.
For Ala-Archa and Chu, however, this is caused by the larger difference between the predictions and performance of the best
20 models compared to the other catchments, as indicated by the difference between the best and mean adj. $R^2$ shown and
listed in Figure 4 and Table 2, respectively. This causes a wider distribution of the residuals of the best 20 models and thus
higher predictive uncertainty. However, if only the best or a smaller selection of the best 20 models are used for a forecast, the
predictive uncertainty would also be reduced. This means, that the uncertainty bands derived depend on the subjective choice
of the number of models to be kept in the model set. Another reason for wider predictive uncertainty bands for later months is
the observed decline in performance during later months in some catchments due to the changed predictor set (e.g. for 6. Chu).
This causes again higher predictive uncertainty bands, which overlay the narrower band from the previous month.

From a formal point of view the uncertainty bands correctly include at least 80% of the observed seasonal discharges, even for
very narrow bands (e.g. in June for 3. Chirchik or 9. Karadarya). This indicates that the uncertainty estimation derived from
the regression residuals provide a reliable estimation of the uncertainty associated to model selection, and can be used to derive
decisions based on the forecasts given by the MLR model sets. However, it must be noted that the derived uncertainty bands
represent the predictive uncertainty of the MLR models fitted to the available time series. They do not account for any
uncertainty stemming from a possible lack of representativeness of the rather short time series used. Longer discharge time
series might show a different variability of seasonal discharge, which would then not be covered by the derived models.
However, as the models can be updated every year in future, this potential problem is expected to decrease with further use of
the approach in the Central Asian Hydromet Services.

Moreover, it has to be noted that the estimated uncertainty cover only the model selection uncertainty. Other uncertainty
sources are:

- model structure, which is assumed to be rather low given the high explained variances;

- data sources, which is not quantifiable, but might be high, particularly the discharge data;

- the performance criteria for selecting the best models.

The last aspect has been tested. Using other performance criteria as PREMS can result in a slightly different selection of best models, but more often just in a different order of the best models. The best PREMS model is not necessarily the best cross validated $R^2$ model, or the best MAE or RMSE model. However, as this mainly affects the ordering of the best models, the results in terms of predictions and predictive uncertainty of the model set, if unweighted as presented, would be very similar.

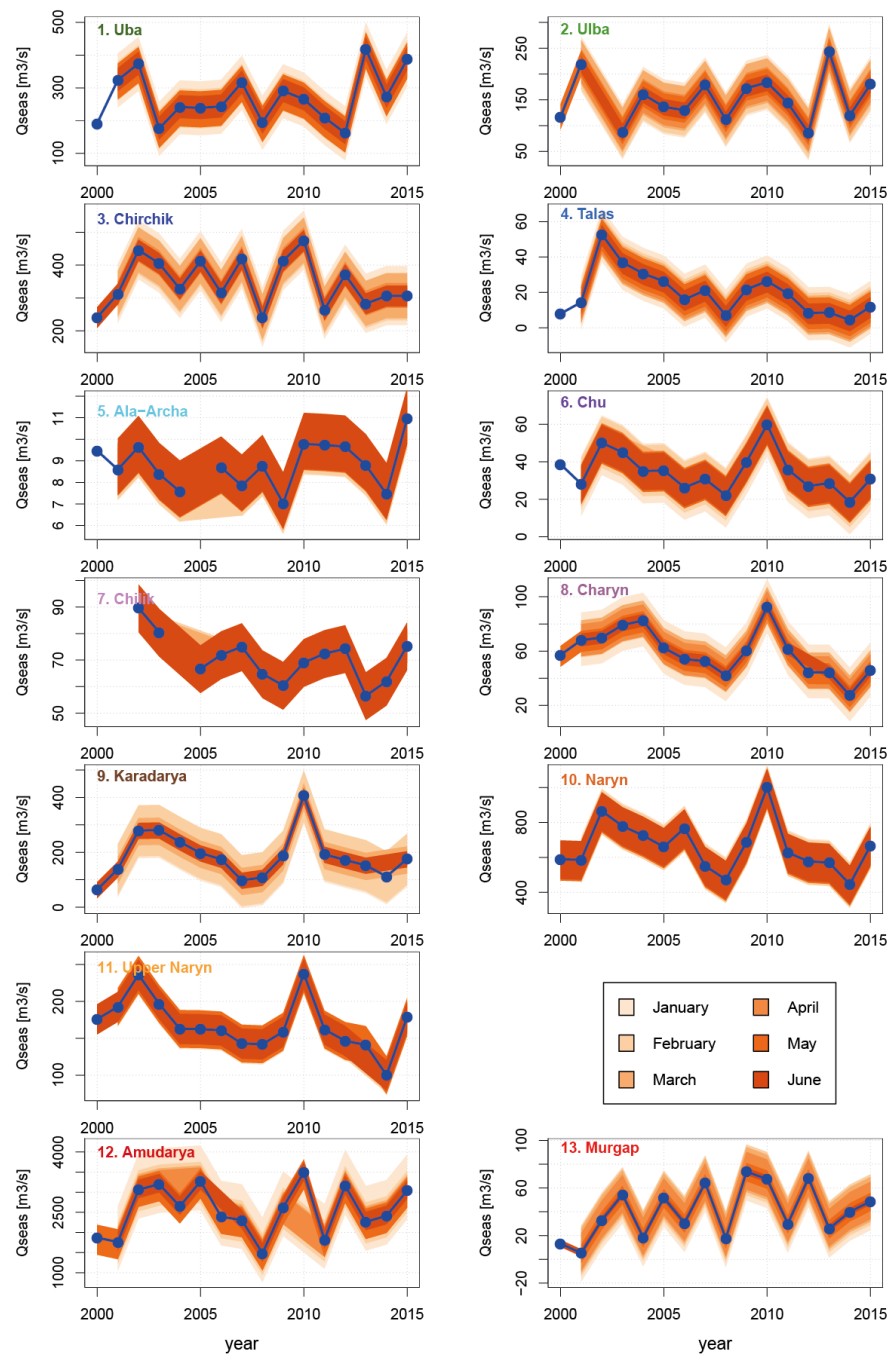

**Figure 6: 80% predictive uncertainty bands for all catchments and forecasts months. The blue lines indicate the observed seasonal discharges.**

In addition to the predictive uncertainty also the reliability of the forecasts was quantified by PIT diagrams and PIT scores. Figure 7 shows the PIT diagrams for every catchment and all forecast months using the forecasts of the selected set of models. The PIT diagrams show that the model set predictions are in most cases close to the 1:1 line, i.e. provide reliable forecasts.

5    However, in some cases the predictive uncertainty is under-estimated (PIT diagram lines with pronounced vertical component around the 50% quantile). This means that some of the predictive uncertainty bands presented in Figure 6 are too narrow to reliably quantify the predictive uncertainty. This is mostly the case for the late forecasts with high skill, where the models in the set often produce very similar forecasts. In addition to the diagrams a PIT score was calculated as the area between the PIT curve and the 1:1 line as a summarizing indicator for the reliability (Renard et al., 2010). The theoretically least reliable model

10   has a score of 0.5, a perfect model a score of 0. The highest score, i.e. the lowest reliability, of all models is 0.2, with the majority of the models being in the range of 0.07-0.15. Interpreting the scores with the curves in the PIT diagram it can be deduced that the reliability of model sets with PIT scores $\leq 0.1$-0.14 is acceptable. For higher scores the predictive uncertainty derived from the model set is likely to be underestimated.

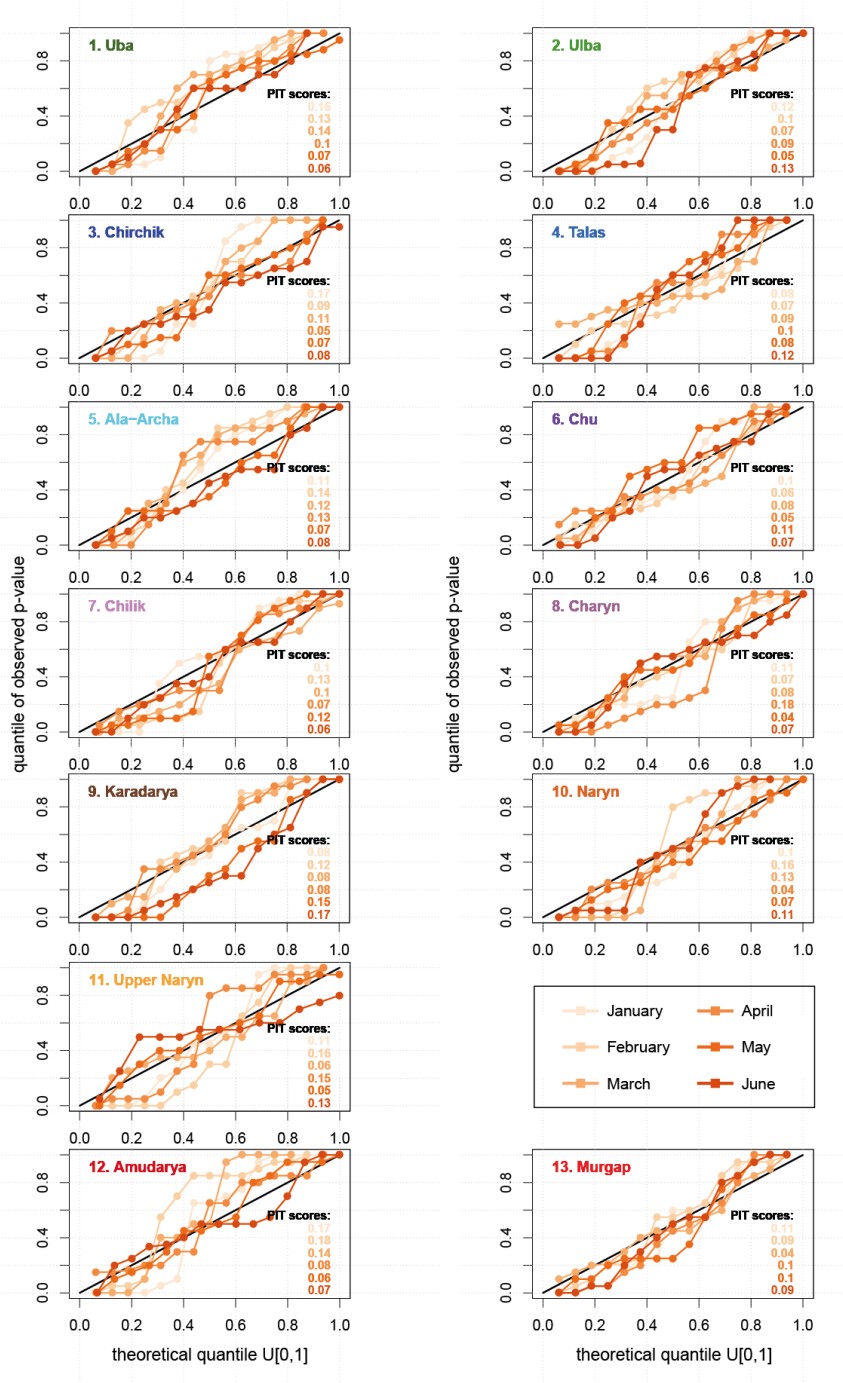

**Figure 7: PIT reliability diagrams for every catchment and forecast month. The PIT score is calculated as the area between the reliability plots and the 1:1 line as suggested in Renard et al. (2010). The lower the PIT score, the higher the reliability. The theoretically least reliability score is 0.5, the best 0. The colour codes of the PIT scores indicate the forecast month as in the legend.**

**4.2 Predictor importance (Is there some hydrological process information in linear models?)**

Figure 8 illustrates the importance of the predictors of the selected MLR models as absolute fractions of the $R^2$ values, whereas it is not differentiated between individual predictors, but rather between predictor classes described in 3.1. The left panel of Figure 8 shows the importance for the single best LOOCV model, while the right panel shows the average importance of the predictors for the best 20 LOOCV models. A comparison of the left and right panels shows that the predictor selection and importance for the different catchments and prediction months of the best model is quite similar to the mean of the best 20 models. This indicates that the predictor selection for the models in the set is quite stable, and hence that the predictor selection is not random, but rather has some hydrological meaning. However, an interpretation of the contributions of the different factors is complicated by the use of the composites, which are almost always selected as one or more predictors in the MLR models. Nevertheless, some general features can be identified from Figure 7:

- Typically there is no single factor dominating the explained variance, with the exception of Karadarya, where the composites have an exceptionally large share on the explained variance. But as the composites are comprised of the other predictors (except antecedent discharge), this statement is actually valid for all catchments. This indicates that the winter snow accumulation providing the bulk of the seasonal discharge is best described by a combination of the factors determining the extent and water equivalent of the snow pack in the catchments (precipitation, temperature, snow coverage). Omitting one of these predictors leads in fact to a reduction in model performance.

- There is a general and plausible trend for higher importance of antecedent discharge in the later prediction months. In this period it can be expected that antecedent discharge has higher predictive power of the seasonal discharge compared to the winter months, i.e. during the accumulation phase, because it directly indicates the magnitude of the discharge generation from snowmelt. This finding is valid for most catchments except Chirchik, Ala-Archa and Chilik. For Chirchik the importance of antecedent discharge is almost constant throughout the prediction months, both for the best model and on average. Contrary to this, antecedent discharge has very little importance for Ala-Archa and Chilik. For Ala-Archa this observation can be explained by the very small catchment size and thus the quick response of discharge to precipitation events and snowmelt, i.e. lower transit times, but also with the high proportion of glacier melt during the summer months. The high importance of precipitation, which is higher than in any other catchment particularly in the later prediction months, also supports this reasoning. For Chirchik and Chilik, however, no plausible explanation can be derived from the basic catchment characteristics presented here.

- The importance of the snow coverage predictors indicate a regional differentiation of the predictor importance. For the two catchments in the Altai region (1. Uba, 2. Ulba, cluster 1 in Figure 3) snow coverage as an individual factor is of less importance compared to the other regions. This observation can be attributed to different snow cover characteristics in these catchments, which have lower altitudes compared to other catchments in this study. Therefore, snow accumulation in these catchments is comparably low and quickly responds to increasing temperature already in

the spring months. Seasonal snow cover variations obtained from the MODSNOW-Tool (Gafurov et al, 2016) for these catchments also illustrate sudden snow cover depletion in the month of April for both catchments, and for Uba with multiple depletions also in winter months until April (analysis not shown in this study). Thus, snowmelt is not important in these catchments for seasonal summer discharge, although it may be of high importance for spring

discharge, which is beyond the focus of this study. The reverse line of argument can be applied for the relatively high importance of snow coverage for the high altitude central Tien Shan catchments (10. Naryn & 11. Upper Naryn) with mean annual temperatures below zero (cf. Table 1), where snow coverage alone explains up to almost 40% of the explained variance by the MLR models, to which the share of snow coverage contained in the composites has to be added. For these catchments snow coverage alone is thus already a good indicator of the seasonal discharge.

•   In terms of predictor importance no obvious differences can be detected on average (right panel in Figure 8) between the Tien Shan and Pamir discharge regimes identified in the cluster analysis (Figure 3), with the exception of the Naryn catchments as stated above. The mean predictor importance figures for those catchments are all very similar. This can indicate that although the variability of seasonal discharges varies with geographical location, the runoff generating processes seem to be similar.

This general interpretation of the predictor importance shows that the selection of the predictors, particularly the change of predictors with prediction months and geographic region, has some hydrological meaning. Due to the simplicity of the approach and the simple linear relationship between the predictors, it is unlikely that more hydrological process information and understanding can be extracted from the MLR results. If this can be achieved at all, then on individual catchment basis

only and by the interpretation of the exact predictors, i.e. not aggregated by classes as above. This is, however, beyond the scope of this study. But nevertheless, the observation described above indicate that the general runoff generation processes can be described by linear models, and that the presented forecast results are unlikely obtained by pure chance only.

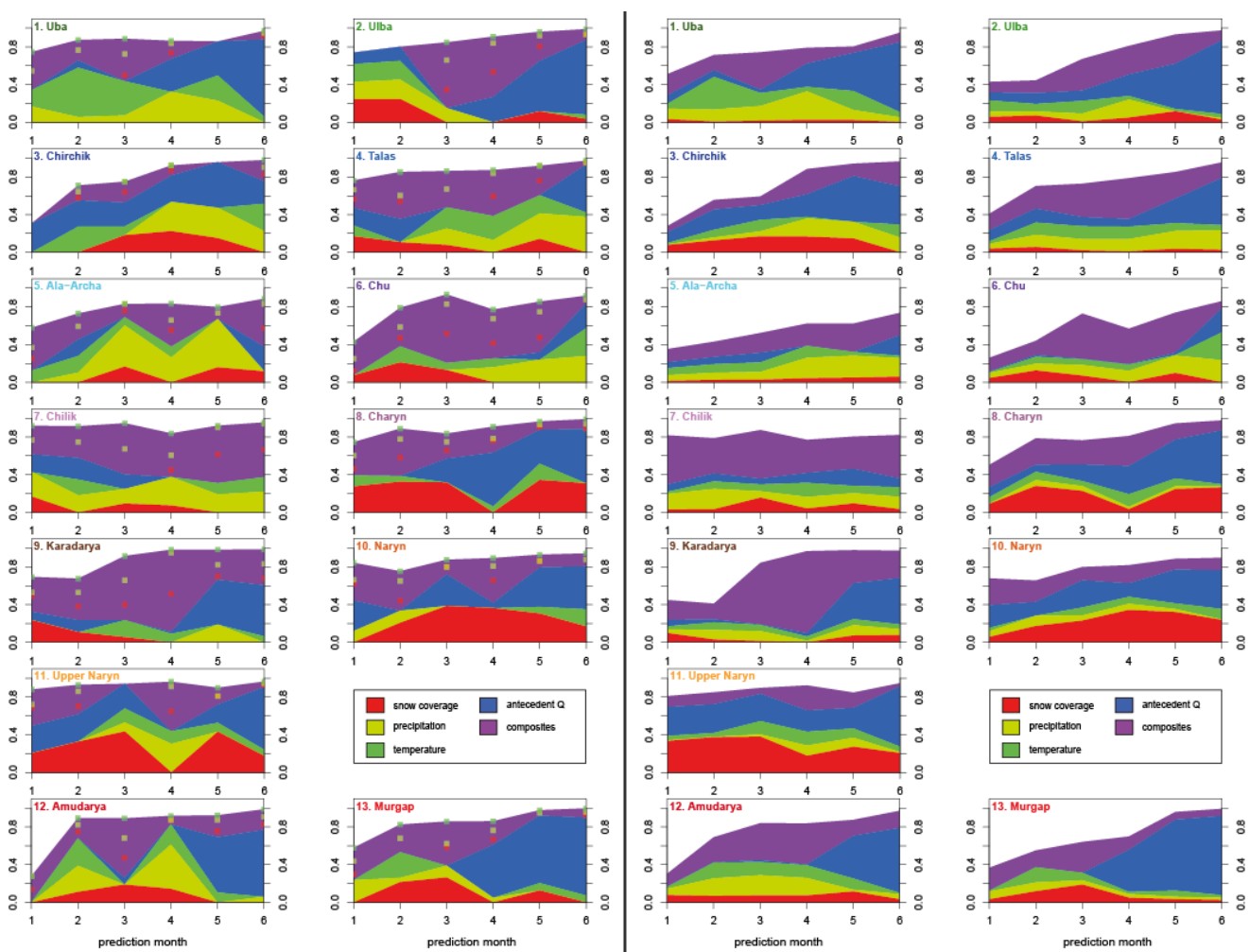

**Figure 8: Importance of the predictors in the linear models as absolute contribution to the explained variance (R²) for all catchments and prediction months. Left: of the best LOOCV model; Right: on average for the best 20 LOOCV models. Squares in the left panel figures indicate the presence of the different predictors used in the composites: snow cover, precipitation and temperature, using the same colour codes as for the individual predictors.**

## 4.3 Potential of operational application

A lot of management and strategic decisions are based on seasonal forecasts of water availability in CA. The main consumer of water resources in the Aral Sea basin is the agricultural sector, which is based on one of the world's largest irrigation systems (Dukhovny and de Schutter, 2011). Very important decisions based on water availability forecasts are the planning of agricultural production crop types and water allocation through the irrigation network. Also the estimation of agricultural yield

is related to water availability and is needed for country income planning, that heavily depends on agricultural export in some countries. Therefore reliable forecasts of seasonal water availability is essential for the economies of Central Asian states.

In order to design a generic and readiliy applicable forecats tool the presented method was designed according to the needs and data availability of the Central Asian Hydromet Services, which are responsible for the seasonal forecasts. The method is

based on station data readiliy available to the state agencies, thus fulfilling a core prerequisite for an operational implementation of the method. Moreover, the procedure for deriving forecast models is fairly simple and implemented in the open source software R. Therefore no limitations due to licence issues exist. The model development is automated requiring only some basic definitions as e.g. the formatting and provision of the predictor data as ASCII text files, and the specification of the prediction month. Therefore the code can be applied by the staff of the Hydromet Services after a short training. However, it

has to be noted that the provided predictor data should be as complete as possible in order to avoid spurios model fitting results (overfitting). Due to the automatic model fitting the algorithm may find best performing models fitted to a few years only, if too many predictor data are missing. The chances of overfitting are then greatly increased as the degree of freedom of the linear models, i.e. the ratio of the years used for fitting to the variables in the prediction models, decreases.

The presented model system can also be run with alternative predictor data. For example, it has been tested using gridded

ERA-Interim re-analysis data for precipitation and temperature, averaged monthly over the individual catchment areas. Similar, if not better results as presented were obtained. However, due to the latency of at least two months until the data is released, an operational use of the model system with ERA-Interim data is not feasible at the moment.

## 6 Discussion and Conclusions

The presented study aimed at the development of a flexible and generic forecast model system for the prediction of the seasonal (April-September) discharge in Central Asian river basins, with the final goal of operational use at the Central Asian Hydromet Services. In order to achieve this the data requirements were kept as low as possible, using only monthly precipitation and temperature data from a single station in the individual catchments, accompanied by operationally processed monthly MODIS snow coverage data and monthly antecedent discharge. Based on this core predictor data set, a variety of monthly, multi-

monthly and composite predictors were automatically derived for different prediction dates. The predictors were then used for predicting the seasonal discharge with Multiple Linear Regression models (MLR). In order to avoid overfitting, restrictions were set on the selection and number of predictors in each MLR, and the models were tested for significance and for robustness by a Leave-One-Out Cross Validation (LOOCV). A set of significant prediction models was then selected based on the Predictive Residual Error Mean of Squares (PREMS) of the LOOCV.

The prediction model system was tested for the period 2000 – 2015 on a selection of 13 different river basins in different geographic and climatic regions, and with different catchment characteristics. It could be shown that the models provided good to excellent predictions for all catchments and for all defined prediction dates, resp. lead times. For the first prediction on

January 1$^{st}$, i.e. for a lead time of three months, the explained variance (expressed as adjusted R$^2$) is already high in the range of 0.46 – 0.86 for 9 catchments. For the following prediction on February 1$^{st}$ the explained variance is above 0.59 for all catchments, and increases further with the following months. For the important prediction date for the planning of water resources in the region on April 1$^{st}$ just before the high flow season, adj. R$^2$ values of the best models for each catchment are

in the range 0.68 – 0.97, indicating exceptional high performance for a seasonal forecast.

The automatic selection of the predictors and their importance revealed some geographic or temporal patterns. Geographically the northern Altai catchments differ in the predictor selection of the best LOOCV-MLR models from the other regions as snowmelt in this region has less contribution to seasonal discharge (April – September), with snow cover often reduced to zero already in early spring. For all catchments the importance of antecedent discharge is increasing with progressing prediction

dates. This is plausible from a hydrological perspective: While during the winter months the discharge is dominated by groundwater contribution, the discharge in April and later contains information about the snowmelt process, and has thus predictive power. This means in summary that the selected predictors and their importance have some hydrological meaning, thus supporting the validity of the forecast models derived by the model system. However, it has to be noted that specific features of runoff generation in the catchments cannot be detected and discovered by the rather abstract level of predictors,

predictor importance and the very basic catchment characteristics presented here. Overall, the presented simple forecast system proved to be able to provide robust, very skilful, and reliable forecast models for Central Asia.

The reason for the high performance is surely the temporal separation of most of the annual precipitation (snow in winter), and the runoff generation (snowmelt in spring and summer).The forecast is thus based on an estimation of the snow pack accumulation in winter and its snow water equivalent, for which the predictors and their combinations provide proxy

information. Moreover, the proxy information is not forecasted, but measured, thus providing more reliable information compared to forecasted predictors.. An additional incorporation of climate predictions from dynamical (Kim et al., 2012) and statistical (Gerlitz et al. 2016) seasonal prediction models is unlikely to further increase the forecast skill. The contribution of spring and summer rainfall to discharge variations appears to be dispensable, as represented by the high forecast performance of the applied predictor variables. Furthermore, the prediction of seasonal climate anomalies is highly uncertain, particularly

for the rather dry summer season, which impedes its application for seasonal runoff forecasts (Gerlitz et al. 2016). Potential improvements could be achieved using gravity based water storage variations, as e.g. provided by the GRACE mission. The total water storage variation monitored by GRACE should actually map the snow accumulation and the snow water content over the whole winter period. This information could be used as predictor for catchments large enough to match the spatial resolution of GRACE.

As the timely separation of precipitation and runoff is a unifying feature of all Central Asian headwater catchments encompassing high-mountain ranges, the model system is able to perform exceptionally well for all tested catchments. It is thus also very likely, that the model system will also work well in the Central Asian catchments not included in this study, with some limitations for very small catchments. Thus, the proposed methodology provides a generic and flexible tool for the development of seasonal discharge forecast models for Central Asian rivers. This tool can be used by the responsible Hydromet

Services without the need for larger investments in hardware, software, and education and training of staff. In fact, the model system is already tested in four Central Asian national Hydromet Services. The only prerequisite for the application of the model system is the availability of meteorological data from stations within the catchments. If this is not given, stations nearby can be used as an alternative. However, they need to be representative for the meteorological variability in the catchment. As this study has shown, this might work e.g. for stations located downstream of the discharge stations within the same but larger catchment. Moreover, a comparable forecast skill can be expected in other regions with similar climatic settings, e.g. the South American dry Andes or the Western U.S. (e.g. the Sierra Nevada). The provided information of seasonal water availability could also be used in dam operation and dam safety procedures, and strategic flood hazard management plans.

## Acknowledgements

This work was undertaken within the frame of the CAWa project (www.cawa-project.net) funded by the German Federal Foreign Office as part of the German Water Initiative for Central Asia (grant number AA7090002).

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

**Annex**

**Annex 1: Correlation of seasonal discharge to sub-seasonal discharge**

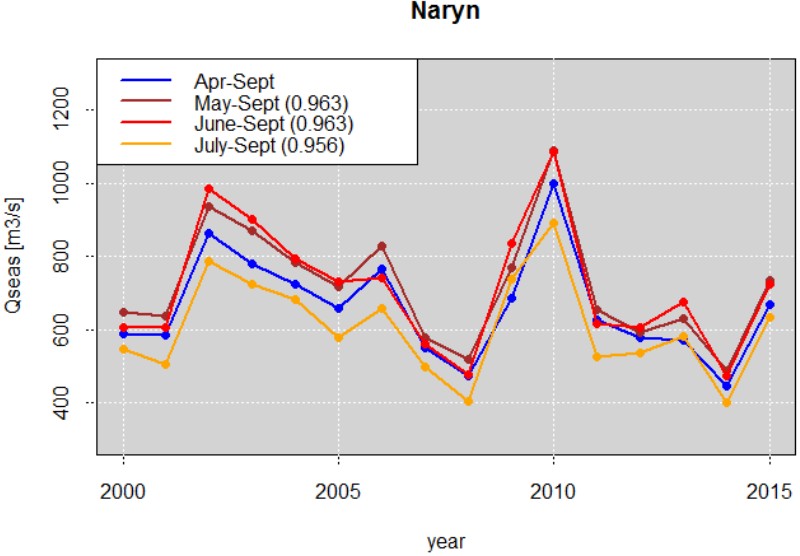

**Figure A1: Comparison of seasonal discharge for the whole vegetation period April to September to sub-seasonal discharge time series taking the Naryn basin as example. The sub-seasonal series are highly correlated to the seasonal time series. Numbers in the legend provide the linear correlation coefficient of the sub-seasonal discharges to the seasonal discharge of the whole vegetation period.**

**Annex 2: Predictors used for the different prediction dates**

The following paragraphs list the predictors created and used for the different forecasts dates, ranging from January 1st to June 1st. The predictors are abbreviated, with *snowcov* and *sc* denoting the snow coverage in the catchment derived by the MODSNOW-tool, *precip* the station records of precipitation, *temp* the station records of temperature, *Q* the discharge recorded at the river gauges. Catchment characteristics and the locations of the gauges are listed in Table 1. The data for all predictors are monthly values (mean for snow coverage, temperature and discharge, total for precipitation), with *jan* indicating January values, *feb* February values, *mar* March values, *apr* April values, *may* May values and *jun* June values.

Multi-monthly values are mean values of the monthly values spanning over several months, whereas the range of the months included is indicated by the concatenation of the indicators of the months, e.g. *janapr* means multi-monthly means for the period January to April, or *febmar* indicates the mean of the months February and March. The predictor abbreviations are combined with the indicators for the months. *snowcov_apr* thus stands for the mean snow coverage of the catchment in April, or *precip_janmar* for the mean of the monthly total precipitation for the months January to March.

For the composites the predictors included are listed by their abbreviations, followed by the indicators for the months. For calculating the composites, the monthly values of the predictors denoted by the month indicators are multiplied. E.g.

*sc_temp_mar* thus means the product of the mean snow cover in March and the mean temperature in March, or *sc_temp_precip_janmay* denotes the product of the multi-monthly means January to May of snow coverage, temperature and precipitation.

**Predictors used for prediction on January 1st**

Snowcover:

  snowcov_dec, snowcov_nov, snowcov_oct, snowcov_octdec

Precipitation:

  precip_dec, precip_nov, precip_oct, precip_novdec, precip_octdec

Temperature:

  temp_dec, temp_nov, temp_oct, temp_novdec, temp_octdec

Composites snowcover x temperature:

  sc_temp_octdec

Composites snowcover, x precipitation:

sc_precip_octdec

Composites temperature x precipitation:

  temp_precip_dec, temp_precip_nov, temp_precip_oct, temp_precip_octdec

Composites snowcover, x temperature x precipitation:

  sc_temp_precip_octdec

Antecedent discharge:

  Q_dec, Q_nov, Q_oct, Q_novdec, Q_octdec

**Predictors used for prediction on February 1st**

Snowcover:

snowcov_jan, snowcov_dec, snowcov_nov, snowcov_oct, snowcov_octjan

Precipitation:

  precip_jan, precip_dec, precip_nov, precip_oct, precip_decjan, precip_novjan, precip_octjan

Temperature:

  temp_jan, temp_dec, temp_nov, temp_oct, temp_decjan, temp_novjan, temp_octjan, sc_temp_jan

Composites snowcover, x temperature:

  sc_temp_jan

Composites snowcover x precipitation:

  sc_precip_jan

Composites temperature x precipitation:

temp_precip_jan, temp_precip_dec, temp_precip_nov, temp_precip_oct, temp_precip_decjan, temp_precip_novjan, temp_precip_octjan

Composites snowcover x temperature x precipitation:

sc_temp_precip_octjan

Antecedent discharge:

Q_jan, Q_dec, Q_nov, Q_oct, Q_decjan, Q_novjan, Q_octjan

**Predictors used for prediction on March 1st**

Snowcover:

snowcov_feb, snowcov_jan, snowcov_janfeb, snowcov_dec, snowcov_nov, snowcov_oct, snowcov_octfeb

Precipitation:

precip_feb, precip_jan, precip_dec, precip_nov, precip_oct, precip_janfeb, precip_decfeb, precip_novfeb, precip_octfeb

Temperature:

temp_feb, temp_jan, temp_dec, temp_nov, temp_oct, temp_janfeb, temp_decfeb, temp_novfeb, temp_octfeb

Composites snowcover x temperature:

sc_temp_jan, sc_temp_feb, sc_temp_janfeb

Composites snowcover x precipitation:

sc_precip_jan, sc_precip_feb, sc_precip_janfeb

Composites temperature x precipitation:

temp_precip_jan, temp_precip_feb, temp_precip_dec, temp_precip_nov, temp_precip_oct, temp_precip_janfeb, temp_precip_novfeb, temp_precip_octfeb

Composites snowcover x temperature x precipitation:

sc_temp_precip_janfeb, sc_temp_precip_octfeb

Antecedent discharge:

Q_feb, Q_jan, Q_dec, Q_nov, Q_oct, Q_janfeb, Q_decfeb, Q_novfeb, Q_octfeb

**Predictors used for prediction on April 1st**

Snowcover:

snowcov_mar, snowcov_feb, snowcov_jan, snowcov_janmar, snowcov_febmar

Precipitation:

precip_mar, precip_feb, precip_jan, precip_dec, precip_nov, precip_oct, precip_febmar, precip_janmar, precip_decmar, precip_novmar, precip_octmar

Temperature:

temp_mar, temp_feb, temp_jan, temp_dec, temp_nov, temp_oct, temp_febmar, temp_janmar, temp_decmar, temp_novmar, temp_octmar

Composites snowcover x temperature:

sc_temp_mar, sc_temp_febmar, sc_temp_janmar

Composites snowcover x precipitation:

sc_precip_mar, sc_precip_febmar, sc_precip_janmar, sc_precip_mar_decmar, sc_precip_mar_novmar

Composites temperature x precipitation:

temp_precip_jan, temp_precip_feb, temp_precip_mar, temp_precip_febmar, temp_precip_janmar, temp_precip_decmar, temp_precip_novmar

Composites snowcover x temperature x precipitation:

sc_temp_precip_mar, sc_temp_precip_febmar, sc_temp_precip_janmar

Antecedent discharge:

Q_mar, Q_feb, Q_jan, Q_dec, Q_nov, Q_oct, Q_febmar, Q_janmar, Q_decmar, Q_novmar, Q_octmar

**Predictors used for prediction on May 1st**

Snowcover:

snowcov_apr, snowcov_mar, snowcov_feb, snowcov_janapr, snowcov_febapr, snowcov_marapr

Precipitation:

precip_apr, precip_mar, precip_feb, precip_jan, precip_marapr, precip_febapr, precip_janapr, precip_decapr, precip_novapr,

precip_octapr

Temperature:

temp_apr, temp_mar, temp_feb, temp_jan, temp_marapr, temp_febapr, temp_janapr, temp_decapr, temp_novapr, temp_octapr

Composites snowcover x temperature:

sc_temp_mar, sc_temp_apr, sc_temp_marapr, sc_temp_febapr

Composites snowcover x precipitation:

sc_precip_mar, sc_precip_apr, sc_precip_marapr, sc_precip_febapr

Composites temperature x precipitation:

temp_precip_jan, temp_precip_feb, temp_precip_mar, temp_precip_apr, temp_precip_febapr, temp_precip_marapr,

temp_precip_octapr

Composites snowcover x temperature x precipitation:

sc_temp_precip_mar, sc_temp_precip_apr, sc_temp_precip_marapr, sc_temp_precip_janapr

Antecedent discharge:

Q_apr, Q_mar, Q_feb, Q_jan, Q_marapr, Q_febapr, Q_janapr, Q_decapr, Q_novapr, Q_octapr

**Predictors used for prediction on June 1[st]**

Snowcover:

snowcov_apr, snowcov_mar, snowcov_feb, snowcov_janapr, snowcov_febapr, snowcov_marapr

Precipitation:

precip_may, precip_apr, precip_mar, precip_feb, precip_jan, precip_aprmay, precip_marmay, precip_febmay, precip_janmay, precip_octmay

Temperature:

temp_may, temp_apr, temp_mar, temp_feb, temp_jan, temp_aprmay, temp_marmay, temp_febmay, temp_janmay, temp_octmay

Composites snowcover x temperature:

sc_temp_mar, sc_temp_apr, sc_temp_marmay

Composites snowcover x precipitation:

sc_precip_mar, sc_precip_apr, sc_precip_marmay

Composites temperature x precipitation:

temp_precip_feb, temp_precip_mar, temp_precip_apr, temp_precip_may, temp_precip_marmay, temp_precip_octmay

Composites snowcover x temperature x precipitation:

sc_temp_precip_mar, sc_temp_precip_apr, sc_temp_precip_marmay, sc_temp_precip_janmay

Antecedent discharge:

Q_may, Q_apr, Q_mar, Q_feb, Q_jan, Q_aprmay, Q_marmay, Q_febmay, Q_janmay Q_octmay

**Annex 3: Autocorrelation of seasonal discharge time series**

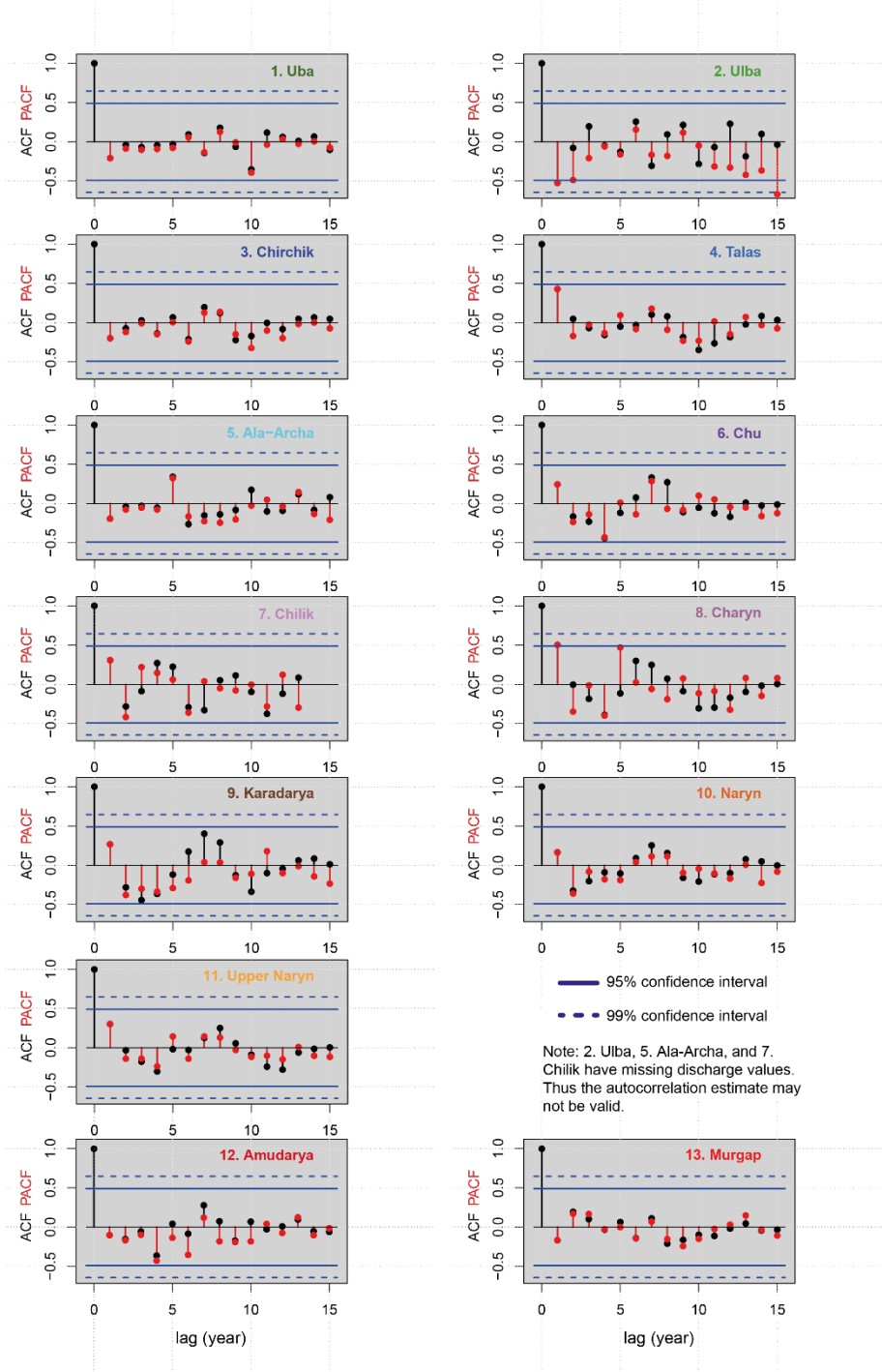

**Figure A2: Auto-correlation (black) and Partial auto-correlation (red) of the seasonal discharge tome series for all catchments and possible lags**

**Annex 4: Formal test for MLR assumptions**

The residuals of the models are tested for normality by the Shapiro-Wilk test for normality. Doing so, one has to bear in mind that this test is based on a sample size of maximal 16 values for each model only, so the test may not provide meaningful results. The table below shows the test result for every model, catchment, and forecast month. A "1" indicates normal distributed residuals, "0" not normal distributed residuals. "NA" indicates that no more models with significant predictors could be found. For every forecast month up to 20 indices are given according to the set of best 20 models to be retained. The table shows that for most of the models (91%) the test was positive, i.e. the residuals are normally distributed, even for this rather low and possibly not representative sample size.

Test for normal distributed residuals, for every catchment, prediction month, and selected 20 models
1 = normal distributed, 0 = not normal distributed, NA = no valid model found

| | January | February | March | April | May | June |
|---|---|---|---|---|---|---|
| Uba | 11111111111111111111 | 10101011011010010110 | 10000111011100001011 | 11111111111111101110 | 11111011011111111111 | 11111111111111111111 |
| Ulba | 11111111111111111111 | 11110111011111011110 | 11101111111111111111 | 11111111111111111111 | 00111111100111110111 | 11111111111111111111 |
| Chirchik | 1111111110101111111NA | 11111111111111111111 | 11111111111111111111 | 11111111111111110100 | 11111111100001111111 | 11111010111111110011 |
| Talas | 11111111110001111111 | 11111111110111111111 | 11111111111111111111 | 11111111111111111111 | 11111111111111111111 | 00111011111011111111 |
| Ala-Archa | 11111011111100111111 | 10110111111111111011 | 11101111111111001111 | 11111111111111111111 | 10111010111111011111 | 11111111111111111111 |
| Chu | 111110111111 NA NA NA NA NA NA NA | 11101111111111111101 | 11111111111111111101 | 11110001101111110011 | 11111111111111111110 | 11111111111110111111 |
| Chilik | 11111111111111111111 | 11111111111111111111 | 10111111011011111111 | 11101111101111011111 | 11100000111010011011 | 11011110111111111111 |
| Charyn | 11111111111111111111 | 11111111111111111111 | 11111111111111111111 | 11111111111111111111 | 11111111111111111111 | 11111111100111111111 |
| Karadarya | 11111111110101111110 | 11111111111111111111 | 11111100011101110110 | 11111111111111110111 | 10111111111111111111 | 01101101111111111111 |
| Naryn | 11111110111111111111 | 11111111111111111111 | 11111111111111111111 | 11111111111111111111 | 11111111111111111111 | 11111111111101111111 |
| Upper Naryn | 11111111111111111111 | 11111011100011111111 | 00101111111111111110 | 11111111111110111111 | 11111111111111111111 | 11111111111111001111 |
| Amudarya | 11111111111 NA NA NA NA NA NA NA | 11111111111111111111 | 11111111111111111111 | 11111101111111111111 | 11111110001011111111 | 11111111111111110001 |
| Murgap | 11111111111111111101 | 11111111111110111111 | 11111111111101111111 | 11111111101110011101 | 11111111111111111111 | 11111111111111111111 |

Furthermore it was tested if the residuals are independent applying a test for autocorrelation with lag 1 at significance level p = 0.05. In the table below a "0" indicates independence, a "1" dependence. It shows that 96% of the models have independent residuals.

Test for autocorrelated (independent) residuals, for every catchment, prediction month, and selected 20 models, lag = 1
1 = correlated, 0 = not correlated, NA = no valid model found

| | January | February | March | April | May | June |
|---|---|---|---|---|---|---|
| Uba | 00000000001001000010 | 10001000000010111000 | 00000000000000000000 | 00000000000000000000 | 00000000000000000000 | 00100000000000000000 |
| Ulba | 10000101000000101001 | 01000011000100000010 | 00000000000000000000 | 00000000000000000000 | 00001000000100000000 | 00000111110000000000 |
| Chirchik | 00000000000000000000NA | 00000000000000000000 | 00000000000000000000 | 00000000000000000000 | 00000000000000000000 | 00000000000000000000 |
| Talas | 00000000000000000000 | 00000000000000000000 | 00000000000000000000 | 00000000000000000000 | 00000000000000000000 | 00000000000000000000 |
| Ala-Archa | 00000000000000000000 | 00000000000000001101 | 00000000000000000000 | 00000000000000000000 | 10000000000000000000 | 00000000000000000000 |
| Chu | 00000000000 NA NA NA NA NA NA NA | 00000000000000001000 | 01100000000000000000 | 00000000000000000000 | 00000000000000000000 | 00000100010000000000 |
| Chilik | 00000000000000000000 | 00000000000000000000 | 00000000000000000000 | 00000000000000000000 | 00000000000000000001 | 00000000000000000000 |
| Charyn | 00000000100000000 | 00000000000000000000 | 00000001000000000000 | 00000000000000000000 | 10000000000000000000 | 00000000000000000000 |
| Karadarya | 01000000010000010000 | 00000000000000000000 | 00000000000000000000 | 00000000000000000000 | 00000000000000000000 | 00000000000000000000 |
| Naryn | 00000000000000000000 | 00000000000000000000 | 00000000000000000000 | 00000000000000000000 | 00000000000000000000 | 00000010000000000000 |
| Upper Naryn | 00000000000000000000 | 00000000000000000000 | 00000100000000000000 | 00000000000000000000 | 00111100000000000000 | 00000011001100011100 |
| Amudarya | 00000000000 NA NA NA NA NA NA NA | 00000000000000000000 | 00000000000000000000 | 00000000000000000010 | 00000010000000000000 | 00001000000000000000 |
| Murgap | 00000000000000000000 | 10001000000000000000 | 00000000000000000000 | 00000010000000000000 | 00000000000000000000 | 00000000000000000000 |

Last the Breusch-Pagan test for heteroscedasticity was applied to the residuals. This test shows that 99.5% of the models have homoscedastic residuals. In the table below a "1" indicates homoscedastic residuals, a "0" heteroscedastic residuals according to the test.

| Test for homoscedastic residuals, for every catchment, prediction month, and selected 20 models | | | | | |
|---|---|---|---|---|---|
| 1 = homoscedasticity test (Breusch-Pagan test) passed, 0 = homoscedasticity test not passed, NA = no valid model found | | | | | |
| | January | February | March | April | May | June |
| Uba | 11111111111111111111 | 11111111111111111111 | 11111111111111111111 | 11111111111111111111 | 11111111111111111111 | 11111111111111111111 |
| Ulba | 11111111111111111111 | 11111111111111111111 | 11111111111111111111 | 11111111111111111111 | 11111111111111111111 | 11111111111111111111 |
| Chirchik | 1111111111111111111 NA | 11111111111111111111 | 11111111111111111111 | 11111111111111111111 | 11111111111111111111 | 11111111111111111111 |
| Talas | 11111111111111111111 | 11111111111111111111 | 11111111111111111111 | 10111111111111011111 | 11111111111111111111 | 11111111111111111111 |
| Ala-Archa | 11111111111111111111 | 11111111111111111111 | 11111111111111111111 | 11111111111111111111 | 11111111111111111111 | 11111111111111111111 |
| Chu | 111111111111 NA NA NA NA NA NA NA NA | 11111111111111111111 | 11111111111111111111 | 11111111111111111111 | 11111101111111111111 | 11111111111111111111 |
| Chilik | 11111111111111111111 | 11111111111111111111 | 11111111111111111111 | 11111111111111111111 | 11111111111111111111 | 11111111111111111111 |
| Charyn | 11111111111111111111 | 11111111111111111111 | 11111111111111111111 | 11111111111111111111 | 11111111111111111111 | 11111111111111111111 |
| Karadarya | 11111111111111111111 | 11111111111111111111 | 11111111111111111111 | 11111111111111111111 | 11111111111111111111 | 11111111111111111111 |
| Naryn | 11111111111111111111 | 11111111111111111111 | 11111111111111111111 | 11111111111111111111 | 11111111111111111110 | 11111111111111111111 |
| Upper Naryn | 11111111111111111111 | 11111111111111111111 | 11111111111111111111 | 11111111111111111111 | 11111111111111111111 | 11111111111111111111 |
| Amudarya | 111111111111 NA NA NA NA NA NA NA NA | 11111111111111111111 | 11111101111111111111 | 11111111111111111111 | 11111111111111111111 | 11111111111111111111 |
| Murgap | 11111111111111111111 | 11111111111111111111 | 11111111111111111111 | 10011111111111111111 | 11111111111111111111 | 11111111111111111111 |

