# Peer review of "Statistical forecast of seasonal discharge in Central Asia using observational records: development of a generic linear modelling tool for operational water resource management"

_Hydrology and Earth System Sciences, 2017_

## Referee Comment (RC1) · Anonymous Referee #1 · 28 Jun 2017

There is an urgent need to improve the safety and operation of impoundments in Central Asia, yet hydrometeorological data to support inflow forecasting and water management are in short supply. This manuscript seeks to address these needs by developing a standard multiple linear regression model of melt season (April-September) discharge in 13 catchments. Forecasts are based on suites of predictors (precipitation, temperature, snow cover and composite variables) in January to June, and tested using a cross-validation technique applied to 16 years of monthly data. Following an exhaustive evaluation of all possible permutations of monthly and averaged predictors,

best-performing models, and 20 near-optimal models are retained. The attendant mix of predictors and uncertainty bounds are then examined for months leading up to and at the start of the forecast season. Variations in forecast skill are qualitatively linked to catchment characteristics.

The overall approach to model development is necessarily pragmatic given the data and technical constraints of the region. Despite the simplicity of the approach, high explained variance ($R^2$) is reported and the authors have bounded forecasts using envelopes of predictor suite uncertainty. However, it is unclear whether the underpinning data comply with the assumptions of the MLR model (i.e. linearity of relationships, homoscedasticity, no outliers, normally distributed and uncorrelated residuals). Furthermore, given the small number of cases (16) and relatively large number of independent variables (4) it is essential that significance levels and adjusted $R^2$ values are reported for all retained MLR models. Significance of the model coefficients should also be tested and any insignificant variables removed. In some models, the predictor variable (e.g. May discharge) is not fully independent of the forecast variable (April to September discharge).

On this basis, publication is recommended subject to the following major revisions, minor corrections and clarifications.

Main comments

[Abstract] Please incorporate more headline results, such as the range of forecast skill for forecasts issued before the onset of the main melt season, as well as typical forecast biases.

[Table 1] Add additional information on the mean annual precipitation, temperature and winter snow cover area in each catchment.

[Section 2.1] Explain the method and purpose of the hierarchical clustering. What metrics were used to compare catchments and to establish cluster membership? The three

clusters should be linked much more explicitly to subsequent discussions of predictor sets (in section 4.2).

[Section 3] How significant are evaporative losses from the catchments and how might this component of the water balance be represented within MLR models?

[Section 3.1] To maintain independence of the predictors, only variables up to March should be used to build models of April-September discharge. After March, the forecast period should be progressively reduced. For instance, predictors between January and April could be used to build models of May-September discharge, January to May variables for June-September discharge, and so forth. Results from models with overlapping predictors and forecast variable should be removed.

[Section 3.2] More rigour is needed in testing for violations of MLR assumptions (i.e. linearity of relationships, homoscedasticity, no outliers, normally distributed and uncorrelated residuals). This could be captured in tabular format with a matrix showing which assumptions (if any) are violated in each catchment.

[Section 3.3] Equations for the lmg algorithm should be provided, and the method of selecting predictors should be described more clearly. Significance of all model coefficients should be formally tested and any insignificant variables removed. All reported $R^2$ values should be adjusted for sample size, and accompanied by a statement of significance. Then, only models that pass the specified level(s) of significance should be retained.

[Table 2] Report only adjusted $R^2$ values for the overall best, and 20 best models. Results for forecasts issued in April, May and June should only cover the periods May-September, June-September and July-September respectively. The legend should be updated accordingly.

[Section 4.1] The discussion of predictive uncertainty should acknowledge other components, including from data quality, choice of model type/ structure, choice of objective

function(s), model parameters. As noted, the uncertainty bands associated with the 20 best models reflect the number of models retained. When more stringent tests of model skill are applied (see comments on section 3.3 above), fewer models may pass. In any event, the criteria for model inclusion within the ensemble used for uncertainty estimation should be stated explicitly.

[Section 4.3] Add a paragraph on the specific operational decisions that are already, or could be, supported by seasonal discharge forecasts in Central Asia.

[Conclusions] Note that seasonal forecasting of precipitation could provide useful information in catchments and years with relatively little winter snowpack accumulation. Seasonal and sub-seasonal forecasts of extreme rainfall could also be important for hazard management (floods, landslides) and dam safety. Note also that the winter precipitation, summer melt situation applies in the Western U.S. too. Add a paragraph on further research opportunities.

Minor corrections and clarifications

[P1, L19] Note that seasonal forecasts can also contribute to improved dam safety.

[P1, L31] State the range of river catchment areas.

[P2, L7] Typo "The Central Asian region..."

[P2, L25] Omit "actually".

[P3, L4] Provide full publication details for the Hydromet Services questionnaire.

[P4, L27] Typo "catchmentss".

[P4, L27] Note that some of the catchments are nested (i.e. not independent) such as the Upper Naryn and Naryn, so the actual sample size is smaller than 13.

[P7, L12] Non sequitur – please clarify why the need for cross-validation and hierarchical clustering follows from the observation that the discharge regimes vary between

catchments.

[P10, L20] Presumably all variables used in composites (e.g. temperature and precipitation) are normalized by their mean and variance such that they have equal weight in the MLR model?

[P11, L4] Provide the equation for the Predicted Residual Error Sum of Squares (PRESS). Note also that had a different objective function been selected, different sets of predictors might have emerged.

[P11, L14] Please clarify "a set of specific models of the best models".

[Figure 4] Improve legibility by removing the grey background from each panel. Avoid use of red with green lines as these will be indistinguishable for some readers.

[Table 3] Explain how the number of "good" forecasts can be higher for the mean than for the best model in some catchments (e.g. Uba, January).

[P19, L22] Please clarify "possible lack of representativeness of the time series used for the "real" variability of the seasonal discharge in Central Asia".

[P21, L7] Please clarify the sentence "This indicates that the predictor selection..."

[Figure 7] Ideally the presentation and discussion of the predictors would be organized by the three clusters described in section 2.1.

[P24, L3] Typo "precipiutation".

[P24, L25] Report only adjusted R2 values with accompanying significance level(s).

---

## Short Comment (SC1) · 28 Jun 2017

Comments

Figure 1: Some gauges are located downstream of impoundments (e.g. catchment 12, Amu Darya). Are the data used corrected for management of upstream reservoirs or does management impact the flow record? A figure showing the annual regime could help to depict whether flows are natural or managed.

Table 2: Adjusted R2 values may be more suitable to report due to small sample sizes

Table 2: Model performance could be benchmarked against the long term average or persistence forecast to quantify additional skill provided by MLR models

General: Winter hydropower production is also a key use of water in the region as well as irrigation provision. Comment might be made as to whether these models could be useful for hydropower planning as well as summer irrigation demands.

General: The inclusion of local stakeholders in the authorship adds significant insight into the paper. This could be enhanced via the authors commenting on how the forecasts presented here facilitate improved water management in the region, possibly providing examples of better decisions made possible by the forecasts. Furthermore, insight could be provided regarding if the forecasts produced here fulfil the requirements of hydromet agencies, or if there are any specific areas in which the models do not perform satisfactorily requiring further research.

Minor corrections

P6, L11: Typo - capitalised while

P7, L3: States "continuous time series for all data and stations were available" when later it is stated that there is some missing data (e.g. Figure 2)

P11, L27: "Figures presented in 4.3" – should this be 4.2?

P19, L19 and P21, L12: Catchment 9 is referred to as Andijan rather than Karadarya.

Figure 7: Possibly label x-axis as Jan, Feb, etc. rather than 1-6 to ease interpretation

General: Inconsistent spelling of Murgab/Murgap, e.g. Table 1 and Figure 1

---

## Referee Comment (RC2) · Anonymous Referee #2 · 18 Jul 2017

This paper proposes to use standard multiple linear regression (MLR) to predict season streamflow for 13 catchments in Central Asia. The predictors are antecedent precipitation, streamflow, temperature, and snow depth. The different combinations of predictors are tested using MLR under the framework of leave-one-out cross validation (LOOCV) and using the metric of predicted residual error sum of squares (PRESS). At the end, "the best 20 forecast models" are picked out for the prediction of future streamflow. In general, the paper is well-written and the results are clearly presented. In the meantime, there are comments for further improvements of the paper:

[Figure]

First of all, it is widely known that the predictability of seasonal streamflow is generally from two sources, i.e., catchment storage and future climate [Hamlet and Lettenmaier, 1999; Chiew and MacMahon, 2002; Wood et al., 2002; Schepen et al., 2012; Crochemore et al., 2017]. However, in this paper, the predictors of future climate, which can be atmospheric circulation indices and GCM/RCM outputs, are not considered at all. That is to say, this paper only accounts for the predictability from catchment storage. As a result, the forecasts as are presented in this paper are not deemed "best" and they can be further improved. The authors are encouraged to consider circulation indices in seasonal streamflow forecasting. It is noted that NOAA provides a collection of more than 30 climatic indices (https://www.esrl.noaa.gov/psd/data/climateindices/list/).

Second, the analysis of predictive uncertainty is too simple to be informative in this paper. It is pointed out that for ensemble and probabilistic forecasts, the attributes of reliability and skill are of key importance [Murphy, 1993, What Is a Good Forecast? An Essay on the Nature of Goodness in Weather Forecasting]. Reliability can be diagnosed using the PIT reliability diagram or PIT histogram [e.g., Wang et al., 2009; Crochemore et al., 2017]. Meanwhile, Skill can be measured using the continuous ranked probability score (CRPS), which is for both deterministic and ensemble forecasts and is equivalent to the mean absolute error (MAE) for deterministic forecasts [Hersbach, 2000]. In addition to the illustrative plots of predictive uncertainty, the authors are encouraged to perform a comprehensive examination of forecast reliability and skill.

There are also some minor comments: 1. As for LOOCV, it can lead to artificial overestimation of forecast skill if the streamflow series exhibit strong auto-correlation. It is worthwhile to check the serial autocorrelation of streamflow. Or, a more rigorous leave-five-years-out cross validation (L5OCV) ought to be applied. 2. In terms of predictors of catchment storage, the use of multi-monthly means as the predictor values is sensible. 3. The paper suggests to use the "the best 20 forecast models". This setting is empirical and it is rare in peer studies. Please clarify why.

References: http://ascelibrary.org/doi/abs/10.1061/%28ASCE%290733-9496%281999%29125%3A6%28333%29 http://www.tandfonline.com/doi/abs/10.1080/02626660209492950 http://onlinelibrary.wiley.com/doi/10.1029/2001JD000659/abstract http://journals.ametsoc.org/doi/abs/10.1175/JCLI-D-11-00156.1 http://www.hydrol-earth-syst-sci.net/21/1573/2017/hess-21-1573-2017.pdf http://journals.ametsoc.org/doi/abs/10.1175/1520-0434%281993%29008%3C0281%3AWIAGFA%3E2.0.CO%3B2 http://onlinelibrary.wiley.com/doi/10.1029/2008WR007355/full http://journals.ametsoc.org/doi/abs/10.1175/1520-0434(2000)015%3C0559:DOTCRP%3E2.0.CO;2

---

## Author Response (AR1)

Dear Dr. Pechlivanidis,

Thank you very much for the kind editor decision and letter. We have now thoroughly revised the manuscript, taking up all the suggestions from the reviewers. As requested, we explained the suitability of the MLR for the forecast in more detail based on hydrological processes. Additionally we included the formal tests of the MLR assumptions in the manuscript, partly in the main text, partly in the Annex. We also included the reliability analysis as suggested by reviewer 2 in the manuscript. Due to the additions the manuscript is now longer compared to the original submission, but we believe still concise enough for a scientific article.

I hope that we have met your and the reviewers requirements, and looking forward to the review results of the revised manuscript.

Kind regards,

Heiko Apel

On behalf of all co-authors.

**Reply to reviewer comment hess-2017-340-RC1**

Heiko Apel[1], Zharkinay Abdykerimova[2], Marina Agalhanova[3], Azamat Baimaganbetov[4], Nadejda Gavrilenko[5], Lars Gerlitz[1], Olga Kalashnikova[6], Katy Unger-Shayesteh[1], Sergiy Vorogushyn[1], Abror Gafurov[1]

[1]GFZ German Research Centre for Geoscience, Section 5.4 Hydrology, Potsdam, Germany

[2]Hydro-Meteorological Service of Kyrgyzstan, Bishkek, Kyrgyzstan

[3]Hydro-Meteorological Service of Turkmenistan, Ashgabat, Turkmenistan

[4]Hydro-Meteorological Service of Kazakhstan, Almaty, Kazakhstan

[5]Hydro-Meteorological Service of Uzbekistan, Tashkent, Uzbekistan

[6]CAIAG Central Asian Institute for Applied Geoscience, Bishkek, Kyrgyzstan

*Correspondence to*: Heiko Apel (heiko.apel@gfz-potsdam.de)

**General referee comment:**

There is an urgent need to improve the safety and operation of impoundments in Central Asia, yet hydrometeorological data to support inflow forecasting and water management are in short supply. This manuscript seeks to address these needs by developing a standard multiple linear regression model of melt season (April-September) discharge in 13 catchments. Forecasts are based on suites of predictors (precipitation, temperature, snow cover and composite variables) in January to June, and tested using a cross-validation technique applied to 16 years of monthly data. Following an exhaustive evaluation of all possible permutations of monthly and averaged predictors, best-performing models, and 20 near-optimal models are retained. The attendant mix of predictors and uncertainty bounds are then examined for months leading up to and at the start of the forecast season. Variations in forecast skill are qualitatively linked to catchment characteristics.

The overall approach to model development is necessarily pragmatic given the data and technical constraints of the region. Despite the simplicity of the approach, high explained variance ($R^2$) is reported and the authors have bounded forecasts using envelopes of predictor suite uncertainty. However, it is unclear whether the underpinning data comply with the assumptions of the MLR model (i.e. linearity of relationships, homoscedasticity, no outliers, normally distributed and uncorrelated residuals). Furthermore, given the small number of cases (16) and relatively large number of independent variables (4) it is essential that significance levels and adjusted $R^2$ values are reported for all retained MLR models. Significance of the model coefficients should also be tested and any insignificant variables removed. In some models, the predictor variable (e.g. May discharge) is not fully independent of the forecast variable (April to September discharge).

On this basis, publication is recommended subject to the following major revisions, minor corrections and clarifications.

We thank the referee for the critical and constructive comments. We provide detailed answers and justifications below, were the main comments are listed.

**Main comments**

[Abstract] Please incorporate more headline results, such as the range of forecast skill for forecasts issued before the onset of the main melt season, as well as typical forecast biases.

The suggestion has been taken up and the abstract reads now as follows:

5   The semi-arid regions of Central Asia crucially depend on the water resources supplied by the mountainous areas of the Tien Shan, Pamir and Altai mountains. During the summer months the snow and glacier melt dominated river discharge originating in the mountains provides the main water resource available for agricultural production, but also for storage in reservoirs for energy generation during the winter months. Thus a reliable seasonal forecast of the water resources is crucial for a sustainable management and planning of water resources. In fact, seasonal forecasts are mandatory tasks of all national hydro-
10   meteorological services in the region. In order to support the operational seasonal forecast procedures of hydro-meteorological services, this study aims at the development of a generic tool for deriving statistical forecast models of seasonal river discharge. The generic model is kept as simple as possible in order to be driven by available meteorological and hydrological data, and be applicable for all catchments in the region. As snowmelt dominates summer runoff, the main meteorological predictors for the forecast models are monthly values of winter precipitation and temperature, satellite based snow cover data and antecedent
15   discharge. This basic predictor set was further extended by multi-monthly means of the individual predictors, as well as composites of the predictors. Forecast models are derived based on these predictors as linear combinations of up to 3 or 4 predictors. A user selectable number of best models is extracted automatically by the developed model fitting algorithm, which includes a test for robustness by a leave-one-out cross validation. Based on the cross validation the predictive uncertainty was quantified for every prediction model. Forecasts of the mean seasonal discharge of the period April to September are derived
20   every month starting from January until June. The application of the model for several catchments in Central Asia - ranging from small to the largest rivers (240 km$^2$ to 290,000 km$^2$ catchment area)– for the period 2000-2015 provided skilful forecasts for most catchments already in January with adjusted R$^2$ values of the best model in the range of $0.3 - 0.8$. The skill of the prediction increased every following month, i.e. with reduced lead time, with adjusted R$^2$ values usually in the range $0.8 - 0.9$ for the best and $0.7 - 0.8$ for the ensemble mean in April just before the prediction period. The later forecasts in May and June
25   improve further due to the high predictive power of the discharge in the first 2 months of the snow melt period. The improved skill of the model ensemble with decreasing lead time resulted in very narrow predictive uncertainty bands at the beginning of the snow melt period. In summary, the proposed generic automatic forecast model development tool provides robust predictions for seasonal water availability in Central Asia, which will be tested against the official forecasts in the upcoming years, with the vision of operational implementation.

[Table 1] Add additional information on the mean annual precipitation, temperature and winter snow cover area in each catchment.

Thanks for the suggestion. We will extend Table 1 as shown below.

**Table 1: List of the catchments for which prediction models are derived with discharge (Q) and meteorological gauging stations used for the prediction. Note that Charvak, Andijan and Toktogul are reservoir inflows summing several tributary inflows. For the Charvak reservoir the mean temperature and precipitation data of three meteo stations located in the catchment was used. Latitude and longitudes are in decimal degrees (WGS84). Q mean seasonal is multiannual mean seasonal discharge from April to September for the period 2000-2015. Mean annual P ist the mean annual precipitation sum of the meteo station for the period 2000-2015. Mean annual T is the mean annual mean temperature of the meteo station for the period 2000-2015. Mean winter SC is the mean of the mean daily snow coverage of January to February for the period 2000-2015.**

| | catchment | discharge station | Q deg. lat | Q deg. long | meteo station | meteo deg. lat | meteo deg. long | meteo altitude [m] | catchment area [km²] | Q mean seas. [m³/s] | mean altitude [m] | mean ann. P [mm] | mean ann. T [°C] | mean winter SC [%] |
|---|---|---|---|---|---|---|---|---|---|---|---|---|---|---|
| 1 | Uba | Shemonaikha | 50.620 | 81.880 | Shemonaikha | 50.620 | 81.880 | 300 | 9324 | 269.2 | 740 | 460 | 3.6 | 69.2 |
| 2 | Ulba | Perevalochnaya | 50.033 | 82.843 | Oskemen | 50.030 | 82.700 | 375 | 5080 | 151.4 | 950 | 483 | 3.8 | 87.7 |
| 3 | Chirchik | Charvak | 41.626 | 69.969 | Chatkal | 41.822 | 71.097 | 2300 | 10903 | 346.21 | 2575 | 708 | 5.5 | 97.3 |
| | | | | | Oygaing | 42.000 | 70.633 | 1620 | 10903 | | | | | |
| | | | | | Pskem | 41.861 | 70.384 | 2220 | 10903 | | | | | |
| 4 | Talas | Kluchevka | 42.581 | 71.836 | Kyzyl-Adyr | 42.616 | 71.586 | 1764 | 6663 | 19.62 | 2424 | 327 | 9.0 | 72.1 |
| 5 | Ala-Archa | Kashka-Suu | 42.650 | 74.500 | Baytik | 42.670 | 74.630 | 1579 | 239 | 8.83 | 3288 | 559 | 3.2 | 79.6 |
| 6 | Chu | Kochkor | 42.250 | 75.833 | Kara Kuzhur | 41.930 | 76.300 | 855 | 4961 | 34.53 | 2934 | 253 | 1.1 | 59.4 |
| 7 | Chilik | Malybai | 43.494 | 78.392 | Shelek | 43.597 | 78.249 | 600 | 3964 | 70.67 | 2603 | 274 | 11.0 | 74.5 |
| 8 | Charyn | Sarytogai | 43.553 | 79.293 | Zhalanash | 43.043 | 78.642 | 1690 | 7921 | 59.06 | 2260 | 507 | 6.1 | 82.4 |
| 9 | Karadarya | Andijan | 40.814 | 73.257 | Ak-Terek | 40.365 | 74.222 | 1190 | 11670 | 186.21 | 2663 | 913 | 9.5 | 82.4 |
| 10 | Naryn | Toktogul | 41.760 | 72.750 | Naryn city | 41.460 | 75.850 | 2040 | 51926 | 653.13 | 2850 | 374 | -5.8 | 88.0 |
| 11 | Upper Naryn | Naryn city | 41.460 | 75.85 | Tien Shan | 41.910 | 78.210 | 3614 | 10343 | 168.64 | 3546 | 345 | -5.8 | 91.0 |
| 12 | Amudarya | Kerki | 37.842 | 65.23 | Kerki | 37.842 | 65.230 | 237 | 287714 | 2551.02 | 2578 | 173 | 17.9 | 56.7 |
| 13 | Murgab | Takhta Bazar | 35.966 | 62.907 | Takhta Bazar | 35.966 | 62.907 | 354 | 35767 | 40.13 | 1707 | 217 | 18.2 | 37.5 |

[Section 2.1] Explain the method and purpose of the hierarchical clustering. What metrics were used to compare catchments and to establish cluster membership? The three clusters should be linked much more explicitly to subsequent discussions of predictor sets (in section 4.2).

We want to show that the different catchments show some differences in the inter-annual variability of the seasonal discharge. This is important, because if all catchment would have the same inter-annual variability, the discharge could theoretically be equally well forecasted with meteorological variables from other catchments with the same variability. This would mean in turn, that the presented ability of the approach to predict the seasonal discharge for the selection of different catchments would provide no additional evidence for the suitability of the approach as a single test case. Cluster memberships were established based on the dissimilarities of the correlation between the seasonal discharge time series of the different catchments, i.e.

basically on the similarity/dissimilarity of the variability of the seasonal discharge as shown in Figure 2. The cluster algorithm starts by assigning a single cluster for each catchments, and starts to reduce the number of clusters by joining the most similar clusters. For the construction of the clusters the Ward algorithm was chosen, which minimizes the variability within the clusters and maximizes the variability between the clusters. This is a standard procedure. Details on this can be found in any statistical textbook.

[Section 3] How significant are evaporative losses from the catchments and how might this component of the water balance be represented within MLR models?

The catchments presented are all mountainous catchment with a cold climate and fast flowing rivers. The evaporative losses from the rivers are thus expected to be low, and do not substantially influence the seasonal discharge to be predicted. Evaporative losses from reservoirs are more likely, but all catchments in the study are without reservoirs (except the Nurek dam in Amudarya, whos influence is negligible in this large catchment. See also reply to short comment SC1), or represent inflows into reservoirs. And in general, evaporative losses are difficult to observe directly and thus to include in the MLR models. Moreover, we do not believe past evaporative losses would have a high predictive power for future discharge. Evaporation is strongly related to radiation/temperature and past temperature is already included as a potential predictor into the MLR models.

[Section 3.1] To maintain independence of the predictors, only variables up to March should be used to build models of April-September discharge. After March, the forecast period should be progressively reduced. For instance, predictors between January and April could be used to build models of May-September discharge, January to May variables for June-September discharge, and so forth. Results from models with overlapping predictors and forecast variable should be removed.

We agree, in order to guarantee independence of the predictors from the predictand this would be the appropriate procedure enabling a fair comparison of the skill of the forecasts before and during the vegetation period. But this is not the purpose of the presented study. We rather aim at providing the best possible forecasts with the given data at hand. As shown in the results and discussion, the observed discharge values (antecedent discharge predictors) from the start of the vegetation period have a high predictive power for the whole vegetation period. Therefore these should be used for the prediction, particularly when a possible application in operational forecast is considered. Besides this, the results would very likely not change much, because the seasonal discharge for April to September is highly correlated to the seasonal discharges for shorter periods. The following figure shows this exemplarily for the Naryn basin. The shorter seasonal mean discharges are very similar to the whole vegetation period April to September, and are highly correlated. The numbers in the legend show the linear correlation coefficient. All correlations are highly significant (p-values $< 10^{-8}$). This means that the performance of models predicting only the discharge ahead is pretty much identical to the presented performance.

[Figure]

Moreover, the presented approach is in line with the official forecast procedures in the Central Asian hydromet services. In order to obtain acceptance of the proposed method in the services and their use in the official forecast procedures it is advisable to follow the prescribed procedures. It is required from the Hydromet Services to issue updated (corrected) forecasts, which include the entire vegetation period (April-September), The water regulation procedures and e.g. agricultural yield estimation are traditionally based on bulk numbers for the entire period. If these procedures are not followed, the obtained results, which are better than the forecasts issued with the existing procedures, might not be implemented and come into practise, and thus a chance would be missed to bring research results into application.

[Section 3.2] More rigour is needed in testing for violations of MLR assumptions (i.e. linearity of relationships, homoscedasticity, no outliers, normally distributed and uncorrelated residuals). This could be captured in tabular format with a matrix showing which assumptions (if any) are violated in each catchment.

The general answer to this comment is: no, the discharge generation in the catchments is not linear, particular if all relevant processes are considered. This has been shown in many hydrological studies. However, this does not mean that linear models cannot be applied. In fact, runoff generation can be approximated by linear models. This has been proven by the many hydrological modelling studies based on linear concepts, e.g. linear storage models. Moreover, hydrological processes can be even better approximated on longer time scales, or on larger spatial scales. This is the basis for the still wide spread use of linear regression in (seasonal) forecast studies (Seibert et al., 2017;Delbart et al., 2015;Dixon and Wilby, 2015). Furthermore,

if the processes to be described show significant non-linear features, using linear models will result in low(er) performance. Predictions and model performance cannot be improved by linear models if processes are non-linear. We thus argue, that the use of linear regression for seasonal forecasts as presented is justifiable by these general considerations, and is actually supported by the good results obtained.

5    However, we also tested the MLR assumptions as suggested by the reviewer in order to show that our general argument holds, i.e. that the seasonal runoff generation in Central Asia can be approximated with linear models.

First we tested if the residuals of the models are normally distributed with the Shapiro-Wilk test for normality. Doing so, one has to bear in mind that this test is based on a sample size of maximal 16 values for each model only, so the test may not provide meaningful results. The table below shows the test result for every model, catchment, and forecast month. Note that

10    the test was performed for a new set of models, where models with insignificant predictors were removed (cf. comment below). A "1" indicates a normal distributed residuals, "0" not normal distributed residuals. "NA" indicates that no more models with significant predictors could be found. For every forecast month up to 20 indices are given. The table shows that for most of the models (89.5%) the test was positive, i.e. the residuals are normally distributed, even for this rather low and possibly not representative sample size.

| Test for normal distributed residuals, for every catchment, prediction month, and selected 20 models | | | | | |
|---|---|---|---|---|---|
| 1 = normal distributed, 0 = not normal distributed, NA = no valid model found | | | | | |
| | January | February | March | April | May | June |
| Uba | 11111111111111111111 | 10101011011010010110 | 10000111011100001011 | 11111111111111101110 | 11111111111111111111 | 11111111111111111111 |
| Ulba | 11111111111111111111 | 11101011011111011110 | 11101111111111111111 | 11111111111111111111 | 11111111111000010000 | 11111111111111111111 |
| Chirchik | 1111111110101111111NA | 11111111111111111111 | 11111111111111111111 | 11111111111111110100 | 00000111100000000001 | 00111001110111111111 |
| Talas | 11111111110001111111 | 11111111110111111111 | 11111111111111111111 | 11111111111111111111 | 11111111111111111001 | 11001111101111111001 |
| Ala-Archa | 11111011111100111111 | 10110111111111111011 | 11101111111111001111 | 11111111111111111111 | 11101101011111110111 | 11111111111111111111 |
| Chu | 1111011111111NANANANANANANA | 11101111111111111111 | 11111111111111111101 | 11110001101111110011 | 11111111111111011111 | 11111111111111111111 |
| Chilik | 11111111111111111111 | 11111111111111111111 | 10111111011011111111 | 11101111101111011111 | 11101010110111011100 | 11111111101001110101 |
| Charyn | 11111111111111111111 | 11111111111111111111 | 11111111111111111111 | 11111111111111111111 | 11111111111111111111 | 11111111111111111111 |
| Karadarya | 11111111111010111110 | 11111111111111111111 | 11111100001101100110 | 11111111111111110111 | 11111111111111111111 | 11101110011001110100 |
| Naryn | 11111110111111111111 | 11111111111111111111 | 11111111111111111111 | 11111111111111111111 | 11100111110111111111 | 11111111111011111111 |
| Upper Naryn | 11111111111111111111 | 11111101110001111111 | 00101111111111111110 | 11111111110111011111 | 11111100001111111111 | 11111111111111111111 |
| Amudarya | 11111111111NANANANANANANA | 11111111111111111111 | 11111111111111111111 | 11111101111111111111 | 11111111101111011110 | 11111111111111111111 |
| Murgap | 11111111111111111101 | 11111111111111011111 | 11111111111101111111 | 11111111101110011101 | 11111111110111101111 | 11111101001111101101 |

Next we tested if the residuals are independent applying a test for autocorrelation with lag 1 at significance level p = 0.05. In the table below a "0" indicates independence, a "1" dependence. It shows that 95.8% of the models have independent residuals.

| Test for autocorrelated (independent) residuals, for every catchment, prediction month, and selected 20 models, lag = 1 | | | | | |
|---|---|---|---|---|---|
| 1 = correlated, 0 = not correlated, NA = no valid model found | | | | | |
| | January | February | March | April | May | June |
| Uba | 00000000001001000010 | 10001000000010111000 | 00000000000000000000 | 00000000000000000000 | 00000000000000000000 | 00000000000000000000 |
| Ulba | 10010101000000101001 | 01000011000100000010 | 00000000000000000000 | 00000000000000000000 | 00000000000000000000 | 11110101110010000000 |
| Chirchik | 0000000000000000000NA | 00000000000000000000 | 00000000000000000000 | 00000000000000000000 | 00000000000000000000 | 00000000000000000000 |
| Talas | 00000000000000000000 | 00000000000000000000 | 00000000000000000000 | 00000000000000000000 | 00000000000000000000 | 00000000000000000000 |
| Ala-Archa | 00000000000000000000 | 00000000000001101 | 00000000000000000000 | 00000000000000000000 | 00000000001000000 | 00000000000000000000 |
| Chu | 00000000000000NANANANANANANA | 00000000000000000000 | 01100000000000000000 | 00000000000000000000 | 00000000000000000000 | 00001100000000100000 |
| Chilik | 00000000000000000000 | 00000000000000000000 | 00000000000000000000 | 00000000000000000000 | 10001000000000000000 | 00000000000000000000 |
| Charyn | 00000000000100000000 | 00000000000000000000 | 00000000000000000000 | 00000000000000000000 | 00000000000000000000 | 00000000000000000100 |
| Karadarya | 01000000010000010000 | 00000000000000000000 | 00000000000000000000 | 00000000000000000000 | 00000000000000000000 | 00000000000000000000 |
| Naryn | 00000000000000000000 | 00000000000000000000 | 00000000000000000000 | 00000000000000000000 | 00110000000000000000 | 00000000000000010000 |
| Upper Naryn | 00000000000000000000 | 00000100100000000000 | 00000000000000000000 | 00000000000000000000 | 00000011110000000000 | 11000000011000001010 |
| Amudarya | 0000000000000NANANANANANANA | 00000000000000000000 | 00000000000000000000 | 00000000000000000010 | 00000000001000000000 | 00000000010000000000 |
| Murgap | 00000000000000000000 | 10001000000000000000 | 00000000000000000000 | 00000001000000000000 | 10000000000000000000 | 00000000000000000000 |

Furthermore we applied the Breusch-Pagan test for heteroscedasticity. This test shows that 99.5% of the models have

20    homoscedastic residuals.

| Test for homoscedastic residuals, for every catchment, prediction month, and selected 20 models | | | | | |
|---|---|---|---|---|---|
| 1 = homoscedasticity test (Breusch-Pagan test) passed, 0 = homoscedasticity test not passed, NA = no valid model found | | | | | |
| | January | February | March | April | May | June |
| Uba | 11111111111111111111 | 11111111111111111111 | 11111111111111111111 | 11111111111111111111 | 11111111111111111111 | 11111111111111111111 |
| Ulba | 11111111111111111111 | 11111111111111111111 | 11111111111111111111 | 11111111111111111111 | 11111111111111111111 | 11111111111111111111 |
| Chirchik | 1111111111111111111NA | 11111111111111111111 | 11111111111111111111 | 11111111111111111111 | 11111111111111111111 | 11111111111111111111 |
| Talas | 11111111111111111111 | 11111111111111111111 | 11111111111111111111 | 10111111111111011111 | 11111111111111111111 | 11111111111111111111 |
| Ala-Archa | 11111111111111111111 | 11111111111111111111 | 11111111111111111111 | 11111111111111111111 | 11111111111111111111 | 11111111111111111111 |
| Chu | 111111111111NANANANANANANA | 11111111111111111111 | 11111111111111111111 | 11111111111111111111 | 11111111111111111111 | 11111111111111111111 |
| Chilik | 11111111111111111111 | 11111111111111111111 | 11111111111111111111 | 11111111111111111111 | 11111111111111111111 | 11111111111111111111 |
| Charyn | 11111111111111111111 | 11111111111111111111 | 11111111111111111111 | 11111111111111111111 | 11111111111111111111 | 11111111111111111111 |
| Karadarya | 11111111111111111111 | 11111111111111111111 | 11111111111111111111 | 11111111111111111111 | 11111111111111111111 | 11111111111111111111 |
| Naryn | 11111111111111111111 | 11111111111111111111 | 11111111111111111111 | 11111111111111111111 | 11111111111001111111 | 11111111111111111111 |
| Upper Naryn | 11111111111111111111 | 11111111111111111111 | 11111111111111111111 | 11111111111111111111 | 11111111111111111111 | 11111111111111111111 |
| Amudarya | 111111111111NANANANANANANA | 11111111111111111111 | 111111011111111111111 | 11111111111111111111 | 11111111111111111111 | 11111111111111011111 |

In summary, we believe that the provided arguments and tests provide sufficient reason and arguments for the use of MLR models for the seasonal forecasts in Central Asia. The tables shown above can be included in an appendix to the manuscript.

[Section 3.3] Equations for the lmg algorithm should be provided, and the method of selecting predictors should be described more clearly. Significance of all model coefficients should be formally tested and any insignificant variables removed. All reported R2 values should be adjusted for sample size, and accompanied by a statement of significance. Then, only models that pass the specified level(s) of significance should be retained.

This comment refers to several section of the manuscript, not only section 3.3. The whole process of predictor selection and model fitting and model selection is described in sections 3.1 and 3.2. Section 3.3 describes the procedure of calculating the predictor importance, which is not relevant for model and predictor selection, but is rather a help for interpreting the selected models and their predictors. We answer to the different points referring to the different sections.

Section 3.1: The selection of the predictors used in the MLR models is described in this section. Additionally tables providing the selected predictors for each forecast month are given in the appendix. We actually think that this is clearly described, and do not see how to improve the description further. The reviewer comment does not provide guidance for this, while the second reviewer seems to be satisfied with our explanations. Therefor we will leave this section as it is, unless more information about what is unclear is provided.

Section 3.2: First, we did not report the significant levels in section 4 and Table 2, as we thought that it is actually obvious that models with such a high explained variance are highly significant, even for this limited sample size. We indicate the significance levels for the best models in Table 2 below, as well as the lowest significance of the selected 20 models for the mean performances.

However, we did not check the significance of the individual predictors in the models. We thank the reviewer for stressing this point. The model selection process has been modified in a way that only models with all predictors significant at $p = 0.1$ are retained. The selection of the models to be retained is still based on the PRESS value from the LOOCV. However, we weighed the PRESS by the number of years for which forecasts are available in order to reduce possible biases due to missing predictor values (i.e. reduced number of samples). This resembles a Predictive Residual Error Mean Squares. Additionally the performance of the models is now reported in terms of adjusted $R^2$ values, as suggested. This lead to lower performance values mainly for the early forecasts, while the high performance of the late forecasts remain very high. Additionally we added the

Mean Absolute Error MAE (relative to the mean seasonal discharge, just as the RMSE) to the performance plots in Figure 4, as requested by the second reviewer. Figure 4 is updated to the figure below, and Table 2 also reports now the adjusted $R^2$ values of the best LOOCV model and the mean of the selected models, where all predictors are significant at p = 0.1. In Figure 4 the green lines for the PRESS values is replaced by a brown line in order avoid red-green blindness problems, as suggested.

**Table 2: Adjusted $R^2$-values of the best performing prediction models from the LOOCV for all catchments and prediction months. "best" indicates the single best model according to the LOOCV, "mean" indicates the mean percentage over the best 20 models according to the LOOCV. The adjusted $R^2$ values are associated with indicators for significance levels.**

|  |  | January | | February | | March | | April | | May | | June | |
|---|---|---|---|---|---|---|---|---|---|---|---|---|---|
|  |  | best | mean | best | mean | best | mean | best | mean | best | mean | best | mean |
| 1 | **Uba** | 0.678 ++ | 0.511 ++ | 0.824 +++ | 0.714 +++ | 0.842 +++ | 0.743 +++ | 0.811 +++ | 0.790 +++ | 0.823 +++ | 0.804 +++ | 0.959 +++ | 0.951 +++ |
| 2 | **Ulba** | 0.624 o | 0.429 + | 0.714 +++ | 0.444 + | 0.781 +++ | 0.672 ++ | 0.869 +++ | 0.811 +++ | 0.943 +++ | 0.932 +++ | 0.983 +++ | 0.975 +++ |
| 3 | **Chirchik** | 0.253 ++ | 0.278 -- | 0.594 +++ | 0.556 ++ | 0.650 +++ | 0.593 ++ | 0.891 +++ | 0.884 +++ | 0.945 +++ | 0.941 +++ | 0.971 +++ | 0.964 +++ |
| 4 | **Talas** | 0.669 +++ | 0.408 + | 0.794 +++ | 0.703 +++ | 0.808 +++ | 0.728 +++ | 0.823 +++ | 0.787 +++ | 0.886 +++ | 0.852 +++ | 0.961 +++ | 0.954 +++ |
| 5 | **Ala-Archa** | 0.393 + | 0.353 o | 0.597 ++ | 0.431 o | 0.758 +++ | 0.524 + | 0.761 +++ | 0.623 ++ | 0.739 +++ | 0.624 ++ | 0.837 +++ | 0.738 |
| 6 | **Chu** | 0.274 + | 0.260 -- | 0.709 +++ | 0.440 o | 0.903 +++ | 0.729 +++ | 0.680 +++ | 0.569 ++ | 0.800 +++ | 0.740 +++ | 0.887 +++ | 0.862 +++ |
| 7 | **Chilik*** | 0.865 +++ | 0.818 ++ | 0.856 +++ | 0.787 ++ | 0.910 +++ | 0.873 +++ | 0.757 +++ | 0.770 ++ | 0.880 +++ | 0.805 +++ | 0.933 +++ | 0.821 +++ |
| 8 | **Charyn** | 0.643 +++ | 0.503 + | 0.844 +++ | 0.786 +++ | 0.792 +++ | 0.765 +++ | 0.873 +++ | 0.810 +++ | 0.949 +++ | 0.944 +++ | 0.985 +++ | 0.975 +++ |
| 9 | **Karadarya** | 0.573 ++ | 0.449 + | 0.589 +++ | 0.411 ++ | 0.880 +++ | 0.845 +++ | 0.976 +++ | 0.968 +++ | 0.977 +++ | 0.979 +++ | 0.981 +++ | 0.973 +++ |
| 10 | **Naryn** | 0.782 +++ | 0.679 +++ | 0.657 +++ | 0.657 +++ | 0.844 +++ | 0.800 +++ | 0.853 +++ | 0.819 +++ | 0.906 +++ | 0.887 +++ | 0.924 +++ | 0.899 +++ |
| 11 | **Upper Naryn** | 0.832 +++ | 0.810 +++ | 0.898 +++ | 0.850 +++ | 0.916 +++ | 0.897 +++ | 0.947 +++ | 0.923 +++ | 0.858 +++ | 0.847 +++ | 0.950 +++ | 0.947 +++ |
| 12 | **Amudarya** | 0.213 + | 0.304 + | 0.841 +++ | 0.691 +++ | 0.857 +++ | 0.840 +++ | 0.878 +++ | 0.839 +++ | 0.897 +++ | 0.876 +++ | 0.983 +++ | 0.972 +++ |
| 13 | **Murgap** | 0.465 ++ | 0.367 o | 0.757 +++ | 0.551 + | 0.802 +++ | 0.642 ++ | 0.807 +++ | 0.700 ++ | 0.970 +++ | 0.960 +++ | 0.997 +++ | 0.993 +++ |

\* the performance of Chilik is not representative and comparable to the other catchments due to too many missing discharge and predictor data.

Significance p: +++ = 0.01, ++ = 0.05, + = 0.1, o = 0.2, -- = >0.2; for mean the lowest significance of the model ensemble

[Figure]

**Figure 4: Performance of the prediction models for the different catchments and prediction months. Adj. $R^2$ best model is the adjusted $R^2$ of the single best LOOCV model, mean adj. $R^2$ is the mean adj. $R^2$ of the best 20 LOOCV models,**

**min adj. R$^2$ is minimum adj. R$^2$ of the best 20 LOOCV models, robustness is mean LOOCV-adj. R$^2$ of the best 20 models divided by the mean adj. R$^2$, RMSE/MAE norm. is the root mean squared error/mean absolute error of the single best model normalized to mean multi-annual seasonal discharge, mean RMSE/MAE norm is the mean root mean square error/mean absolute error of the best 20 LOOCV models normalized to the multi-annual seasonal discharge; PREMS is the predictive residual sum of squares (PRESS) of the single best model, divided by the number of prediction months.**

Section 3.3: The analysis of predictor importance is performed after the best models are selected, i.e. it has no influence on the predictor and model selection. The *lmg* algorithm calculates how much of the overall explained variance is explained by the individual predictors of the selected models. This is principally performed by re-running the model with a single of the selected predictors and calculating the explained variance. Then the other predictors are added and the gain in explained variance is determined (->sequential R$^2$s). Then the importance of a predictor is given either as percentage of the overall explained variance, or as absolute fraction of explained variance. However, in this procedure the sequence of predictors influences the explained variance. In other words, it matters with which predictor the importance analysis starts. The lmg algorithm tests all predictor orderings and calculates the mean importance of every ordering in order to overcome the problem of predictor ordering in sequential R$^2$s. More details are given in the reference provided. We will add some sentences as above for explanation in the revised manuscript.

[Table 2] Report only adjusted R2 values for the overall best, and 20 best models. Results for forecasts issued in April, May and June should only cover the periods May-September, June-September and July-September respectively. The legend should be updated accordingly.

Adjusted R2 values are now reported in Table 2 and Figure 4 (see above). However, as already explained in an earlier answer, we would keep the current procedure, because of the high correlation of the seasonal and sub-seasonal mean discharge and for better transfer into operational practice in Central Asia.

[Section 4.1] The discussion of predictive uncertainty should acknowledge other components, including from data quality, choice of model type/ structure, choice of objective unction(s), model parameters. As noted, the uncertainty bands associated with the 20 best models reflect the number of models retained. When more stringent tests of model skill are applied (see comments on section 3.3 above), fewer models may pass. In any event, the criteria for model inclusion within the ensemble used for uncertainty estimation should be stated explicitly.

The criteria for model inclusion in the ensemble is as stated the best model performance in the cross validation, i.e. the lowest PREMS (=PRESS divided by the number of years for which forecasts can be made by the individual models) value. However, the number of models for the ensemble is set subjectively to 20. This selection is aiming at obtaining a sufficient number of models for an ensemble evaluation of the forecasts. With the newly set restriction on model selection (only models with significant predictors), a few ensembles, particularly for the January prediction have less than 20 models, because not enough models fulfilling the new selection criteria could be identified. There is actually no rule for the number of ensembles members

applied. We left sufficient amount of freedom for this, in order to enable an expert selection of models by the forecasters of the Central Asian hydromet services. The forecasters have a lot of experience with their catchments, and can decide better which forecasts are valuable for them. The forecasters check every model retained for their performances (qantitatively and qualitatively), and select the models accordingly. This means that in practice less models that the 20 presented in the manuscript might be selected, or even more. Another possible rule for ensemble model selection could be to set a threshold in explained variance for the model. However, due to the high explained variances, the threshold must be very high in order to reduce the ensemble members. A fixed $R^2$ threshold would more likely increase the ensemble members in most cases.

We will explicitly discuss other uncertainty sources. Other uncertainty sources are, as mentioned, model structure, which is rather low given the high explained variances; data sources, which is not quantifiable, but might be high, particularly the discharge data; and performance criteria for selecting the best models. This last aspect has actually been tested, but is not included in the manuscript in order to keep the manuscript concise. Using other performance criteria as PRESS for model selection usually results in slight different selection of best models, and often in a different order of the best models. The best PRESS model is not necessarily the best cross validated $R^2$ model. However, as this mainly affects the ordering of the best models, the results in terms of ensemble predictions, if unweighed as presented, will remain the same. In order to illustrate this, we will some sentences in the manuscript. We could also provide tables with further performance criteria (R2, adj., R2, SSQ, MAE, central Asian performance criteria, all for cross validation and full models) for every selected model, forecast month and catchment in the appendix. This will, however, result in 13 x 6 wide tables, i.e. in quite a long appendix. We would abstain from including this bulk of data into the manuscript, unless the editor explicitly requests this.

[Section 4.3] Add a paragraph on the specific operational decisions that are already, or could be, supported by seasonal discharge forecasts in Central Asia.

A lot of management and strategic decisions are based on seasonal forecasts of water availability in CA. The main consumer of water resources in the Aral Sea basin is the agricultural sector with has one of the world's largest irrigation systems (Dukhovny and de Schutter, 2011). Very important decisions based on water availability forecasts are the planning of agricultural production crop types and water allocation through the irrigation network. Also the estimation of agricultural yield is related to water availability and is needed for country income planning that heavily depends on agricultural export in some countries.

[Conclusions] Note that seasonal forecasting of precipitation could provide useful in-formation in catchments and years with relatively little winter snowpack accumulation. Seasonal and sub-seasonal forecasts of extreme rainfall could also be important for hazard management (floods, landslides) and dam safety. Note also that the winter precipitation, summer melt situation applies in the Western U.S. too. Add a paragraph on further research opportunities.

Thanks for the suggestions. We will discuss the possibility of the forecasts for hazard management on more detail.

However, seasonal forecasts of precipitation in Central Asia are difficult and very uncertain. We have studied this in another publication (Gerlitz et al., 2016). We showed, that winter precipitation amounts are highly related to tropical and extratropical

circulation modes (such as ENSO and NAO) and thus exhibit a certain degree of predictability. In contrast, summer precipitation in Central Asia is usually convective, i.e. is triggered by surface heating and associated atmospheric instability. Summer precipitation sums are composed of few single events (occasionally of high intensity) which, however, are rather randomly distributed and non-predictable.

**Minor corrections and clarifications**

[P1, L19] Note that seasonal forecasts can also contribute to improved dam safety.

10  Thanks for the hint. We will add this in the introduction.

[P1, L31] State the range of river catchment areas.

Will be included.

15  [P2, L7] Typo "The Central Asian region. . ."

Will be corrected.

[P2, L25] Omit "actually".

Will be omitted.

[P3, L4] Provide full publication details for the Hydromet Services questionnaire.

The questionnaire was project internal and not published. This was a prerequisite for participation in the questionnaire. The replies to the questionnaires were rather heterogeneous and were further elaborated and specified in the dialogue with the Central Asian forecasting specialists during a workshop. It would, however, go far beyond the scope of this manuscript to
25  compile and publish the detailed answers of the questionnaires and interviews. The content of the questionnaire would also neither improve the quality of the manuscript, nor change the focus, but rather distract the focus.

[P4, L27] Typo "catchmentss".

Will be corrected.

[P4, L27] Note that some of the catchments are nested (i.e. not independent) such as the Upper Naryn and Naryn, so the actual sample size is smaller than 13.

This comment applies for the Naryn basin only. The catchments were nested in order to analyse if the method also works for high alpine catchment such as Upper Naryn with a high degree of glacierization. We will state this in the revised manuscript.

[P7, L12] Non sequitur – please clarify why the need for cross-validation and hierarchical clustering follows from the observation that the discharge regimes vary between catchments.

Thanks for spotting this. The sentences will be changed in "This plot indicates similar but also different inter-annual variability patterns of the different catchments. In order to distinguish between similar and different inter-annual variabilities cross-correlations of the seasonal discharges are calculated and hierarchically clustered (Figure 3).".

[P10, L20] Presumably all variables used in composites (e.g. temperature and precipitation) are normalized by their mean and variance such that they have equal weight in the MLR model?

No, the variables in the composites were simply multiplied.

[P11, L4] Provide the equation for the Predicted Residual Error Sum of Squares (PRESS). Note also that had a different objective function been selected, different sets of predictors might have emerged.

We will include the following description of PRESS:

"The PRESS residuals are defined as $e_{(i)} = |y_i - \hat{y}_{(i)}|$ where $\hat{y}_{(i)}$ is the regression estimate of $y_i$ based on a regression equation computed leaving out the $i$th observation. The process is repeated for all n observations resulting in:

$$PRESS = \sum_{i=1}^{n} e_{(i)}^2 \qquad \text{"}$$

And yes, a different objective criteria might result in a different set of models, but in most cases usually in a different order of the best models. We commented on this earlier. We argue that PRESS is the most appropriate selection criteria, because a desirable forecast would reduce the residuals to a minimum in cross validation.

Moreover, we now normalized the PRESS by the number of years for which forecasts could be made by the individual models in order to avoid biases caused by missing predictor values. This resembles a Predictive Residual Error Mean of Squares, which we term PREMS.

[P11, L14] Please clarify "a set of specific models of the best models".

This refers to the option of selecting individual models by experts of the catchments, i.e. the responsible forecasters in CA. some models might have acceptable performance criteria, but the temporal dynamics might be not acceptable. Or, some models might show high performance, but have too many missing predictors resulting in a spurious good performance. Such models can be excluded from the ensemble by the forecasters.

[Figure 4] Improve legibility by removing the grey background from each panel. Avoid use of red with green lines as these will be indistinguishable for some readers.

The green line for PRESS (now PREMS) has been replaced be a beige/light brown dashed line. We would actually keep the grey background, because we believe that this supports the legibility and enhances the graphical appearance. Because the other reviewer did not comment on this, we would ask the editor to decide whether the background should be changed or not.

5   [Table 3] Explain how the number of "good" forecasts can be higher for the mean than for the best model in some catchments (e.g. Uba, January).

As mentioned above, the best models are selected according to PRESS. In PRESS the residuals are squared, resulting in a different order and occasionally selection of the set of best models compared to a sorting based on absolute residuals, as the Mean Absolute Error MAE and the CA performance criteria defined in equation (1). This means that the best model according to PRESS is not necessarily the best model according to the CA performance criteria. This can result in more "good" forecasts according to the CA criteria on average for the selected ensemble model compared to the best model.

[P19, L22] Please clarify "possible lack of representativeness of the time series used for the "real" variability of the seasonal discharge in Central Asia".

15  This refers to the limited length of the time series used in this study, which might show a different variability compared to longer time series.

[P21, L7] Please clarify the sentence "This indicates that the predictor selection. . ."

We want to express that a) for the best models similar predictors are selected, i.e. that the predictor selection is not random 20  and follows hydrological principles of runoff generation, and b) that the procedure of predictor selection for the models avoids the selection of correlated predictors from the same group, which could be a problem if the restriction in predictor selection were not set.

We will clarify this in the revised manuscript.

25  [Figure 7] Ideally the presentation and discussion of the predictors would be organized by the three clusters described in section 2.1.

This is a welcome suggestion. We will include references to the clusters in the revised manuscript. Also note that Figure 7 has slightly changed because if the updated model selection (only significant predictors), and because the importance is now quantified as absolute contribution to adjusted $R^2$ values:

[Figure]

Figure 7: Importance of the predictors in the linear models as absolute contribution to the explained variance expressed as adjusted $R^2$ for all catchments and prediction months. Left: of the best LOOCV model; Right: on average for the best 20 LOOCV models. Squares in the left panel figures indicate the presence of the different predictors used in the composites: snow cover, precipitation and temperature, using the same colour codes as for the individual predictors.

[P24, L3] Typo "precipiutation".
Will be corrected.

[P24, L25] Report only adjusted R2 values with accompanying significance level(s).
Will be done.

Heiko Apel[1], Zharkinay Abdykerimova[2], Marina Agalhanova[3], Azamat Baimaganbetov[4], Nadejda Gavrilenko[5], Lars Gerlitz[1], Olga Kalashnikova[6], Katy Unger-Shayesteh[1], Sergiy Vorogushyn[1], Abror Gafurov[1]

[1]GFZ German Research Centre for Geoscience, Section 5.4 Hydrology, Potsdam, Germany

[2]Hydro-Meteorological Service of Kyrgyzstan, Bishkek, Kyrgyzstan

[3]Hydro-Meteorological Service of Turkmenistan, Ashgabat, Turkmenistan

[4]Hydro-Meteorological Service of Kazakhstan, Almaty, Kazakhstan

[5]Hydro-Meteorological Service of Uzbekistan, Tashkent, Uzbekistan

[6]CAIAG Central Asian Institute for Applied Geoscience, Bishkek, Kyrgyzstan

*Correspondence to*: Heiko Apel (heiko.apel@gfz-potsdam.de)

**General referee comment:**

This paper proposes to use standard multiple linear regression (MLR) to predict season streamflow for 13 catchments in Central Asia. The predictors are antecedent precipitation, streamflow, temperature, and snow depth. The different combinations of predictors are tested using MLR under the framework of leave-one-out cross validation (LOOCV) and using the metric of predicted residual error sum of squares (PRESS). At the end, "the best 20 forecast models" are picked out for the prediction of future streamflow. In general, the paper is well-written and the results are clearly presented. In the meantime, there are comments for further improvements of the paper:

First of all, it is widely known that the predictability of seasonal streamflow is generally from two sources, i.e., catchment storage and future climate [Hamlet and Lettenmaier, 1999; Chiew and MacMahon, 2002; Wood et al., 2002; Schepen et al., 2012; Crochemore et al., 2017]. However, in this paper, the predictors of future climate, which can be atmospheric circulation indices and GCM/RCM outputs, are not considered at all. That is to say, this paper only accounts for the predictability from catchment storage. As a result, the forecasts as are presented in this paper are not deemed "best" and they can be further improved. The authors are encouraged to consider circulation indices in seasonal streamflow forecasting. It is noted that NOAA provides a collection of more than 30 climatic indices (https://www.esrl.noaa.gov/psd/data/climateindices/list/).

We thank the reviewer for the constructive comments. We fully agree that the predictability of seasonal streamflow depends on the information about catchment storage and future climate, particularly rainfall. However, in Central Asia much of the discharge stems from snow melt, i.e. the winter accumulation, resp. the precipitation in winter. In the Altai catchments and along the Northern rim of the Tien Shan some additional precipitation occurs during spring and early summer (March-July). This precipitation is eventually considered as observations in the late forecasts presented here. However, reliable information about the spring precipitation in advance could possibly improve the early forecasts. We actually studied the seasonal

predictability of precipitation in Central Asia using NAO, ENSO and EA indices as well as automatically selected seas surface temperature regions as predictors in a preceding paper (Gerlitz et al., 2016). Although some skillful models were obtained for winter precipitation, the variability of the seasonal precipitation in Central Asia was strongly underestimated. Furthermore, summer precipitation in Central Asia is usually convective, i.e. is triggered by surface heating and associated atmospheric

5    instability. Summer precipitation sums are composed of few single events (occasionally of high intensity) which, however, are rather randomly distributed and non-predictable. Therefor we did not include the seasonal forecasts of precipitation in the presented linear models, because no additional gain in performance can be expected. Another (practical) reason for this decision was the envisaged operational use by the CA hydromet services. Using station data as presented is fairly easy for them to include in the operational routines, while using climate indices could pose a mental as well as technical barrier for the

10   staff in the services.

Regarding the term "best" we of course refer to the best forecast obtained with the presented approach and predictors. We do by no means imply that the presented models are the best forecast obtainable in general and will make this clear in the revised manuscript.

15   Second, the analysis of predictive uncertainty is too simple to be informative in this paper. It is pointed out that for ensemble and probabilistic forecasts, the attributes of reliability and skill are of key importance [Murphy, 1993, What Is a Good Forecast? An Essay on the Nature of Goodness in Weather Forecasting]. Reliability can be diagnosed using the PIT reliability diagram or PIT histogram [e.g., Wang et al., 2009; Crochemore et al., 2017]. Meanwhile, Skill can be measured using the continuous ranked probability score (CRPS), which is for both deterministic and ensemble forecasts and is equivalent to the mean absolute

20   error (MAE) for deterministic forecasts [Hersbach, 2000]. In addition to the illustrative plots of predictive uncertainty, the authors are encouraged to perform a comprehensive examination of forecast reliability and skill.

Many thanks for the suggestion. Because we are using deterministic models, we have now evaluated the skill in terms of the MAE and plotted it in Figure 4. The MAE is normalized to the mean seasonal discharge for each basin, just as already shown for the RMSE. The MAE skill is very similar to the RMSE, being in the range of 10%-20% for the January forecasts, and

25   below 10% for the most important April forecast:

[Figure]

**Figure 4: Performance of the prediction models for the different catchments and prediction months. Adj. R² best model is the adjusted R² of the single best LOOCV model, mean adj. R² is the mean adj. R² of the best 20 LOOCV models,**

**min adj. R² is minimum adj. R² of the best 20 LOOCV models, robustness is mean LOOCV-adj. R² of the best 20 models divided by the mean adj. R², RMSE/MAE norm. is the root mean squared error/mean absolute error of the single best model normalized to mean multi-annual seasonal discharge, mean RMSE/MAE norm is the mean root mean square error/mean absolute error of the best 20 LOOCV models normalized to the multi-annual seasonal discharge;**

5 **PREMS is the predictive residual sum of squares (PRESS) of the single best model, divided by the number of prediction months.**

We also evaluated the reliability by means of PIT diagrams, as suggested. The plot below shows the PIT diagrams for every catchment and all forecast months using the prediction of the selected ensemble models. The PIT diagrams show that the model

10 ensemble predictions are in most cases close to the 1:1 line, i.e. provide reliable forecasts. However, in some cases the predictive uncertainty is under-estimated, i.e. the predictive uncertainty bands presented in Figure 6 are too narrow. We further calculated a PIT score as the area between the PIT curve and the 1:1 line as a summarizing indicator for the reliability. The theoretically least reliable model has a score of 0.5, a perfect model a score of 0. The highest score, i.e. the lowest reliability, of all models is 0.2, with the majority of the models being in the range of 0.07-0.15. Interpreting the scores with the curves in

15 the PIT diagram it can be stated that the reliability of the models is good for PIT scores <= 0.1. For higher scores the predictive uncertainty is likely to be underestimated. We will include this analysis in the revised manuscript as suggested by the reviewer, and provide the PIT scores as guidelines for the interpretation of the predictive uncertainty bounds.

[Figure]

**Figure: PIT reliability diagrams for every catchment and forecast month. The PIT score is calculated as the area between the reliability plots and the 1:1 line as suggested in Renard et al. (2010). The lower the PIT score, the higher the reliability. The least reliability score is 0.5, the best 0.**

5   There are also some minor comments:

1. As for LOOCV, it can lead to artificial over-estimation of forecast skill if the streamflow series exhibit strong auto-correlation. It is worthwhile to check the serial autocorrelation of streamflow. Or, a more rigorous leave-five-years-out cross validation (L5OCV) ought to be applied.

We checked the autocorrelation and partial autocorrelation of the streamflow time series and plotted it in the figure below.

10   Hardly any autocorrelation at $p = 0.05$ could be detected. Only for 2. Ulba the partial autocorrelation shows some autocorrelation for lag 1 and 2 just above $p = 0.05$. But in summary for all catchments, it can be stated that autocorrelation does not exist in the discharge time series, and thus the proposed LOOCV is an appropriate validation method. We propose to include the figure below in the appendix of the revised manuscript and include the statement above in the text.

[Figure]

Note: 2. Ulba, 5. Ala-Archa, and 7. Chilik have missing discharge values. Thus the autocorrelation estimate may not be valid.

2. In terms of predictors of catchment storage, the use of multi-monthly means as the predictor values is sensible.

Thanks for the supporting comment.

3. The paper suggests to use the "the best 20 forecast models". This setting is empirical and it is rare in peer studies. Please clarify why.

5 We commented on this already in the reply to reviewer 1, thus we quote the reply here:

The number of models for the ensemble is set subjectively to 20. This selection is aiming at obtaining a sufficient number of models for an ensemble evaluation of the forecasts. With the newly set restriction on model selection (only models with significant predictors), a few ensembles, particularly for the January prediction have less than 20 models, because not enough models fulfilling the new selection criteria could be identified. There is actually no rule for the number of ensembles members

10 applied. We left sufficient amount of freedom for this, in order to enable an expert selection of models by the forecasters of the Central Asian hydromet services. The forecasters have a lot of experience with their catchments, and can decide better which forecasts are valuable for them. The forecasters check every model retained for their performances (quantitatively and qualitatively), and select the models accordingly. This means that in practice fewer models than the 20 presented in the manuscript might be selected, or even more.

15 Another possible rule for ensemble model selection could be a defined threshold of $R^2$ for the model. However, due to the high explained variances, the threshold must be very high in order to reduce the number of ensemble members. A fixed $R^2$ threshold would more likely increase the ensemble members in most cases. The selection of the threshold level would also be subjective.

Heiko Apel[1], Zharkinay Abdykerimova[2], Marina Agalhanova[3], Azamat Baimaganbetov[4], Nadejda Gavrilenko[5], Lars Gerlitz[1], Olga Kalashnikova[6], Katy Unger-Shayesteh[1], Sergiy Vorogushyn[1], Abror Gafurov[1]

[1]GFZ German Research Centre for Geoscience, Section 5.4 Hydrology, Potsdam, Germany

[2]Hydro-Meteorological Service of Kyrgyzstan, Bishkek, Kyrgyzstan

[3]Hydro-Meteorological Service of Turkmenistan, Ashgabat, Turkmenistan

[4]Hydro-Meteorological Service of Kazakhstan, Almaty, Kazakhstan

[5]Hydro-Meteorological Service of Uzbekistan, Tashkent, Uzbekistan

[6]CAIAG Central Asian Institute for Applied Geoscience, Bishkek, Kyrgyzstan

*Correspondence to*: Heiko Apel (heiko.apel@gfz-potsdam.de)

**Short comment:**

Figure 1: Some gauges are located downstream of impoundments (e.g. catchment 12, Amu Darya). Are the data used corrected for management of upstream reservoirs or does management impact the flow record? A figure showing the annual regime could help to depict whether flows are natural or managed.

We used the discharge data provided by the hydromet services, which are not corrected for reservoir management. However, except the large Amudarya catchment all the gauges are located upstream of reservoirs, thus the discharge is not regulated. Within the Amudarya catchment the Nurek dam exists in the Vakhsh river, which is the right headwater tributary forming the Amudarya at the conjunction with the Panj river. The catchment area of the Vakhsh river is 31,415 km² at the outlet. This is about 11% of the whole Amudarya catchment at gauge Kerky. Because the Nurek dam is located upstream of the conjunction, the Amudarya catchment area affected by the dam is less than 10%. Assuming further that the reservoir can manage only a fraction of the total discharge of the Vakhsh river, and that the dynamics of the water retention are further buffered by the seasonal mean discharge spanning six months, it can be assumed that the regulating effect of the Nurek dam on the overall seasonal discharge is rather low. Additionally, the dam is operational since 1980, therefore a discontinuity in the time series 2000-2015 can be ruled out. We thus argue that the anthropogenic influence of the seasonal discharge time series of the Amudarya is negligible for the presented study. We will point this out in the revised manuscript.

Table 2: Adjusted R2 values may be more suitable to report due to small sample sizes

As mentioned in the reply to the comments of reviewer 1, we report adjusted $R^2$ values throughout the revised manuscript. The adjusted $R^2$ values decrease (compared to the $R^2$ values) for the early forecasts, while they remain high for the late forecasts.

Table 2: Model performance could be benchmarked against the long term average or persistence forecast to quantify additional skill provided by MLR models.

This is implicitly done in the $R^2$ values. The coefficient of determination $R^2$ (which is synonymous to explained variance and Nash-Sutcliffe efficiency) benchmark the squared residuals, i.e. the observations minus simulated or forecasted discharges, against the squared differences between observations and the mean observed discharge. In other words, $R^2$ values benchmark the model against the most simple model, which is the mean of the observed time series. Any $R^2$ value above 0 thus indicates an improvement compared to using the mean as predictor. Schaefli and Gupta (2007) nicely illustrate this in their paper about the value of Nash-Sutcliffe efficiency. Therefor we don't see any gain in reporting the mean discharge as reference.

General: Winter hydropower production is also a key use of water in the region as well as irrigation provision. Comment might be made as to whether these models could be useful for hydropower planning as well as summer irrigation demands.

The models are valuable for any planning concerning the use of water resources. This also includes the hydropower generation. Reliable forecasts support the reservoir management by planning the release, resp. the storage of water in the reservoirs for the winter season. However, for reservoir management also international treaties between the riparian states need to be considered. The demands of the upstream and downstream countries are often quite opposite, which is the core problem of water management in Central Asia.

General: The inclusion of local stakeholders in the authorship adds significant insight into the paper. This could be enhanced via the authors commenting on how the forecasts presented here facilitate improved water management in the region, possibly providing examples of better decisions made possible by the forecasts. Furthermore, insight could be provided regarding if the forecasts produced here fulfil the requirements of hydromet agencies, or if there are any specific areas in which the models do not perform satisfactorily requiring further research.

This is an interesting suggestion. The matter is, however, very complex. A detailed analysis would certainly by beyond the scope of this manuscript. Additionally there is the time problem. An encompassing assessment of the model performance, advantages and deficits can only be done after the forecasted vegetation season. i.e. after September. Considering the time required for collecting data, evaluating the forecasts, and the experiences and acceptance in the different hydromet services would likely take several months, certainly exceeding the time schedule for this manuscript. However, this suggestion is very welcome and we will consider to collect and summarize the experience with the models in operational forecasting and publish them additionally, maybe as a short comment.

Minor corrections

P6, L11: Typo - capitalised while

P7, L3: States "continuous time series for all data and stations were available" when later it is stated that there is some missing data (e.g. Figure 2)

P11, L27: "Figures presented in 4.3" – should this be 4.2?

P19, L19 and P21, L12: Catchment 9 is referred to as Andijan rather than Karadarya.

Figure 7: Possibly label x-axis as Jan, Feb, etc. rather than 1-6 to ease interpretation

General: Inconsistent spelling of Murgab/Murgap, e.g. Table 1 and Figure 1

5    Thanks for spotting these errors. We will correct them in the revised manuscript. However, we would keep the x-axis labels as 1-6, in order to avoid overloading the figures. Printed on A4 paper they are already very dense.

References:

10    Schaefli, B., and Gupta, H. V.: Do Nash values have value?, Hydrological Processes, 21, 2075-2080, 2007.

**Revised manuscript with changes marked**

[revised manuscript text omitted]

---

## Author Response (AR2)

Dear Dr. Pechlivanidis,

Thank you very much for organizing the review process and the kind decision letter. We have incorporated the latest suggestion of the reviewers. The most significant change made is the restructuring of the late (May and June) forecasts. In order to ensure the independence of predictand and predictors we followed the suggestion of reviewer 5 3 and yourself to forecast only the remaining season, and added the observed discharge to obtain the full seasonal discharge required by the CA authorities. As argued in our replies to the first reviews, the results hardly change due to the high correlation of the sub-seasonal discharge to the full seasonal discharge. Nevertheless, all the figures and tables were revised to accurately map the latest results.

I hope that we have met your and the reviewers requirements, and looking forward to your reply.

Kind regards,

Heiko Apel

On behalf of all co-authors.

**Reply to comments of reviewer 2**

Heiko Apel[1], Zharkinay Abdykerimova[2], Marina Agalhanova[3], Azamat Baimaganbetov[4], Nadejda Gavrilenko[5], Lars Gerlitz[1], Olga Kalashnikova[6], Katy Unger-Shayesteh[1], Sergiy Vorogushyn[1], Abror Gafurov[1]

[1]GFZ German Research Centre for Geoscience, Section 5.4 Hydrology, Potsdam, Germany

[2]Hydro-Meteorological Service of Kyrgyzstan, Bishkek, Kyrgyzstan

[3]Hydro-Meteorological Service of Turkmenistan, Ashgabat, Turkmenistan

[4]Hydro-Meteorological Service of Kazakhstan, Almaty, Kazakhstan

[5]Hydro-Meteorological Service of Uzbekistan, Tashkent, Uzbekistan

[6]CAIAG Central Asian Institute for Applied Geoscience, Bishkek, Kyrgyzstan

*Correspondence to*: Heiko Apel (heiko.apel@gfz-potsdam.de)

**General referee comment:**

The paper describes streamflow prediction in thirteen river basins in Central Asia using combinations of a range of *observed* predictors. I like the idea that is tested and the overall 'simplicity' of the model (though judging by the number of predictor combinations, I doubt that 'simplicity' is necessarily an appropriate term!). I also like the fact that the model is designed to be implemented operationally by the Central Asian Hydromet services.

The method seems reasonable, and makes sense, but is a little confusing at first because it is not immediately apparent that it is based solely on observational (historical) data. Given the manuscript title, I was expecting to read about streamflow forecasting using climate model outputs of precipitation/temperature, or large-scale climate indices. However, the predictions are based solely on historical data (antecedent meteorological measurements and streamflow). I think the use observational records (and particularly the use of antecedent discharge) needs to be stated more clearly in the abstract and in the methods, because it explains why the forecasting skill is so high.

In fact, I would also recommend changing the title to something more explicit like 'Statistical forecast of seasonal discharge in Central Asia using observational records: development of a generic linear modelling tool for operational water resources management'.

We stated the sole use of observation data in the abstract and the main text. We also changed the title as suggested.

It would be interesting also if the authors could provide some comparison of the skill of their method with other forecasting approaches (including GCM/RCM-based methods) in the same region, over different lead times. Such a comparison would be particularly welcome in the last sections, which provide little perspective from the existing literature.

Comparisons with seasonal flow forecasts for the region is difficult, as publications about this can hardly be found. Available are some seasonal forecasts of precipitation, both dynamical and statistical. Both proved to be very uncertain, particularly for the spring and early summer precipitation, which could potentially improve the early forecasts in January and February. An improvement of the forecasts with these products is thus rather unlikely. We included a statement in the conclusion about this.

Overall, I would suggest making the methodology and choice of data as clear as possible in the text (see minor comments).

I would also suggest re-reading very carefully to improve the language.

We did our best to improve the language and to eliminate errors.

Page 1, L.22. Suggest making this more explicit, e.g. 'the development of a generic tool for deriving statistical forecasts of seasonal river discharge using only observational meteorological and hydrological records. The generic model is kept as simple as possible (i.e., does not include any forecast data)'.

Done.

Page 2.

L.5. 'The improved skill of the model ensemble'. I find this language (model ensemble) a little confusing, because to me it suggests that you are using GCM forecasts..

We changed the term "ensemble" to "set" throughout the manuscript in in order to avoid confusion with GCM ensembles.

L.5. 'very narrow predictive uncertainty bands': --> suggest just 'narrow'

Done.

L.22. Run-on sentence (there are a lot of long sentences here at the start..). Suggest changing 'exposed to the westerly air flows being a …' --> 'exposed to the westerly air flows, which are a…'

Done.

L.23. This what? This spatial variation?

Clarified.

L.23. 'with snow melt to be' --> 'where snowmelt is'

Done.

L.28 'They are also among those river basins with the highest share' --> 'These mountains also have a very high fraction ….'

Done.

L.31. Full stop missing

Done.

L.2. THE vegetation period

Done.

L.7. based on the empirical --> based on empirical

Done.

5  L.21. This paragraph needs an introductory sentence, before listing all the examples. What is the main idea of the paragraph?

The purpose of the paragraph is now motivate by an introduction sentence.

L.34. Predictive skill (no s)

Done.

L.13. Tools for automated image acquisition (no the)

Done.

L.23. 'which areas' --> ', whose combined drainage areas'

15  Done.

L.24. Long sentence. For clarity, I would suggest finishing the first sentence and starting a new one, e.g. "automatic predictor selection. The predictors are based on…"

Done.

L.25. "And additionally leverage by the" should be "and additionally leverage the"

20  Done.

L.27. This statement ("it is argued that … ") needs some references (e.g. Schär et al. 2004, Slater et al. 2017)

This statement is based on theoretical considerations. We state this accordingly.

25  L.2. Add a comma after forecast models

Done.

L.2. They cover --> the catchments cover (better to repeat the object at the start of a sentence)

Done.

L.3. remove 'over catchments at'

30  Done.

L.6. Add a comma after three orders of magnitude

Done.

L.18 'For those catchments stations nearby' --> 'For those catchments, nearby stations'. I hope this is discussed later as a potential limitation.

We come back to this in the discussion.

Table 1: 'ist the mean annual precipitation sum' --> 'is the total mean annual precipitation of the meteorological station…'.
Here and elsewhere in the manuscript, 'total' should be used (instead of 'sum').

Done.

Page 7.

L.4. for the presented study --> for the present study

Done.

L.7. Sentence is too long and needs to be split into 2 or 3 sentences to make sense.

Done.

L.2. For all catchmentS

Done.

L.17. Consider removing thus and 'to a large extent' --> 'The analysis of the inter-annual variability broadly maps the …
differences of the catchments considered in this study'

Done.

L.9. Here you use 'snow melt' in 2 words; elsewhere it is written in one word. Please revise for consistency throughout.

We use snowmelt in throughout the manuscript.

L.11. 'Data about the' --> Snow depth and snow water equivalent data

Done.

L2. 'An indicator about' --> an indicator of

Done.

L11. I think you mean Figure 8?

Yes. Corrected.

L.24. 'the separation of the largest share': unclear what this means; needs rephrasing.

The sentence is rephrased to "The reason for the high performance is surely the temporal separation of most of the annual precipitation (snow in winter), and the runoff generation (snowmelt in spring and summer). Due to this temporal separation there is no need to perform a seasonal forecast of the precipitation for the summer period, which is very difficult and uncertain in Central Asia."

Figures

Figure 4. Prediction month needs to be mentioned in the caption (Jan-June). Grey background (in this figure and the other ones) is a bit distracting – consider removing.

Considered. All figures have now a white background.

Figure 5 is interesting and suggests very high skill. I'm surprised that the skill is shown as time-series rather than scatter plots, as it makes it hard to visualise the biases

The purpose of this figure was to show that the models are able to correctly map the temporal variability of the seasonal discharge. Scatterplots cannot show this.

Figure 8. The text is too small to read, even at full size.

The figure can be plotted in landscape if published, which would increase the font. Increasing the font of the plot in portrait would look awkward, because of a proportional mismatch of the figure and font size.

Annexes

Annex 2. I find this list difficult to read; it would have been clearer in scientific notation.

We inserted commas to separate the predictors. This increases the readability.

**Reply to comments of reviewer 3**

Heiko Apel[1], Zharkinay Abdykerimova[2], Marina Agalhanova[3], Azamat Baimaganbetov[4], Nadejda Gavrilenko[5], Lars Gerlitz[1], Olga Kalashnikova[6], Katy Unger-Shayesteh[1], Sergiy Vorogushyn[1], Abror Gafurov[1]

[1]GFZ German Research Centre for Geoscience, Section 5.4 Hydrology, Potsdam, Germany

[2]Hydro-Meteorological Service of Kyrgyzstan, Bishkek, Kyrgyzstan

[3]Hydro-Meteorological Service of Turkmenistan, Ashgabat, Turkmenistan

[4]Hydro-Meteorological Service of Kazakhstan, Almaty, Kazakhstan

[5]Hydro-Meteorological Service of Uzbekistan, Tashkent, Uzbekistan

[6]CAIAG Central Asian Institute for Applied Geoscience, Bishkek, Kyrgyzstan

*Correspondence to*: Heiko Apel (heiko.apel@gfz-potsdam.de)

**General referee comment:**

According to the answers to reviewers provided by the authors and the improved version of the manuscript, I can state that most of the reviewers' concerns have been worked out and the quality of the manuscript has been distinctly improved. However, there are still some points that need to be addressed before considering it ready for publication.

First of all, the overlapping between predictors and the predictand pointed out in section 3.1 should be better explained from my point of view. It is not acceptable that the predictors and the predictand are not fully independent. I also understand that the system operators need to be given bulk numbers for the entire vegetation period (April – September). An easy way to work this out, in my opinion, would be to distinguish between what is forecasted and which result is given to the system operators. Rather than predicting the entire discharge for the April – September period, the MLR models developed for May 1st and June 1st should aim at predicting the discharge for the remaining of the period. Once this remaining discharge is forecasted, it can be added to the discharge already recorded during the season (discharge in April and discharge in April and May respectively) to obtain the forecasted discharge for April – September season. In this way, the independence between predictors and predictand is respected while the system operators are given the required number.

We followed the suggestion and changed the predictand of the May and June forecasts to sub-seasonal discharges containing only the discharge after the prediction date. The May and June forecast models were all computed again to predict the sub-seasonal discharge only. The observed part of the seasonal discharge is then added to the predicted sub-seasonal discharge in order to obtain values for the full seasonal discharge. This is necessary to fulfil the requirements of the hydromet services. Table 2 and Figure 4 now show the adjusted $R^2$ values for both the full seasonal discharge and the sub-seasonal discharge for

comparison. As argued in our replies to the first reviewer comments, the results do not change substantially by this. Nevertheless, all subsequent figures and tables were updated to show the new results.

5 Furthermore, the reason behind the preliminary predictor list (before significance level calculations) given in section 3 (pages 10 and 11) is not provided. Authors' should explain why they chose these predictors and time lags (expert judgement, data analysis, previous studies, etc.). Moreover, the MLRs which have not passed the significance tests (it is indicated that in some months less than 20 models did it) should be excluded from the following calculations.

We explained the rational behind the predictor selection in more detail. The selection is based on a combination hydrological
10 considerations, preliminary tests, expert knowledge, and practical considerations. And of course, all models that did not pass the significance tests were excluded from the set of prediction models and further calculation. We emphasized this in order to avoid confusion.

Finally, I consider necessary to include the R-squared coefficients computed in May and June for just the remaining period of
15 the vegetation season in Table 2 and Figure 4. They can appear together with the ones calculated for the whole period. I agree on the authors' point of view regarding the current practices. However, providing solely the R-squared coefficient for the whole season may lead to the conclusion that the MLRs' performance increases across the vegetation period, while this increase in the R-squared value may be caused simply by the fact that one or two months are already known and, consequently, the uncertain part decreases. Providing R-squared values for the remining period would be more adequate to rank the quality of
20 the forecasting models and find out which of them should be improved.

As mentioned above, we included the adjusted R2 values for the sub-seasonal forecasts in Table 2 and Figure 4.

Minor concerns:
25 1. Page 12, lines 27-30: I would make a reference to Annex 4 and summarize the results of the tests described here rather in section 4 (page 15, lines 8-14).

We prefer to keep the few sentences about the tests for formal MLR requirements at this locations. Otherwise the results of the tests would not appear in the main manuscript at all.

30 2. Page 13, line 6: Which annex are you referring to?

Annex 2. Corrected in the MS.

[revised manuscript text omitted]